# Single-mitosis dissection of acute and chronic DNA mutagenesis and repair

Paul Adrian Ginno[1], Helena Borgers[1], Christina Ernst [2,6], Anja Schneider [1], Mikaela Behm[1], Sarah J. Aitken [2,3,4], Martin S. Taylor [5] ✉ & Duncan T. Odom [1,2] ✉

How chronic mutational processes and punctuated bursts of DNA damage drive evolution of the cancer genome is poorly understood. Here, we demonstrate a strategy to disentangle and quantify distinct mechanisms underlying genome evolution in single cells, during single mitoses and at single-strand resolution. To distinguish between chronic (reactive oxygen species (ROS)) and acute (ultraviolet light (UV)) mutagenesis, we microfluidically separate pairs of sister cells from the first mitosis following burst UV damage. Strikingly, UV mutations manifest as sister-specific events, revealing mirror-image mutation phasing genome-wide. In contrast, ROS mutagenesis in transcribed regions is reduced strand agnostically. Successive rounds of genome replication over persisting UV damage drives multiallelic variation at CC dinucleotides. Finally, we show that mutation phasing can be resolved to single strands across the entire genome of liver tumors from F1 mice. This strategy can be broadly used to distinguish the contributions of overlapping cancer relevant mutational processes.

Cancers are complex ecosystems of competing clones, developing via idiosyncratic evolution. Tumors often arise after decades of chronic mutational processes, such as oxidative damage[1], and bursts of mutagen exposure[2]. Thus, it is difficult to disentangle mutagenic mechanisms that can trigger transformation in single cells.

Chronic oxidation recurrently damages DNA[3], whereas environmental genotoxins such as ultraviolet light (UV)[4] or nitrosamines[5] can result in bursts of mutagenesis. Mammals have complex repair systems to maintain DNA fidelity, including base excision repair (BER) and nucleotide excision repair (NER)[6]. Although these mechanisms can repair most damage[7,8], fixed mutations eventually accumulate in normal tissues, tumors and cultured cells[2]. How DNA damage can result in mutations is not fully understood.

DNA is chronically challenged by reactive oxygen species (ROS), often forming 8-oxo-7,8-dihydro-2′-deoxyguanosine (8-oxo-G). 8-oxo-G is usually repaired by BER[9] but can result in G > T transversions[9,10]

following DNA replication[1]. ROS creates a background mutation landscape independent of additional mutagenic exposure[10]. However, studying endogenous ROS mutation patterns is challenging, because single oxidative events are rare[11].

Acute DNA damage, such as exposure to UV, can compromise genome integrity. The resulting bulky lesions include cyclobutane pyrimidine dimers or pyrimidine (6-4) pyrimidone photoproducts, which can be repaired by NER. The NER process is guided by genomic context: global NER resolves bulky lesions across the genome, whereas transcription-coupled NER (TCR) is active in transcribed regions[12]. Mutations in the NER protein XRCC1 result in xeroderma pigmentosum[13], where patients are UV sensitive. Our work in mouse liver tumors suggests that lesions caused by acute genotoxic exposure predominantly arise by DNA replication[14,15] result in mutational asymmetry across whole chromosomes. Acute and chronic mutagenic processes operate concurrently, complicating analysis of their individual activities.

[1]German Cancer Research Center (DKFZ), Division of Regulatory Genomics and Cancer Evolution, Heidelberg, Germany. [2]Cancer Research UK - Cambridge Institute, University of Cambridge, Cambridge, UK. [3]MRC Toxicology Unit, University of Cambridge, Cambridge, UK. [4]Department of Histopathology, Cambridge University Hospitals NHS Foundation Trust, Cambridge, UK. [5]MRC Human Genetics Unit, MRC Institute of Genetics and Cancer, University of Edinburgh, Edinburgh, UK. [6]Present address: School of Life Sciences, École Polytechnique Fédérale de Lausanne, Lausanne, Switzerland. ✉e-mail: Martin.Taylor@ed.ac.uk; d.odom@dkfz-heidelberg.de

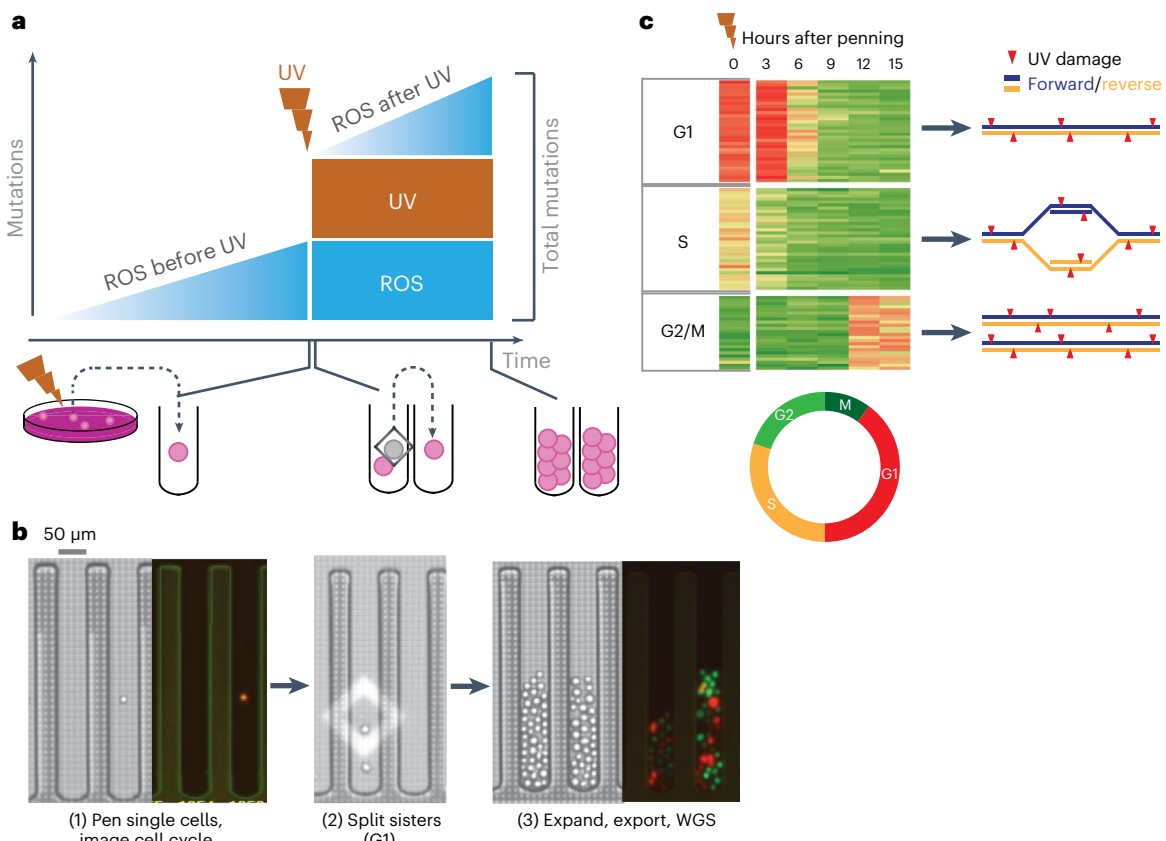

**Fig. 1 | System to distinguish gradual and acute mutational processes in mammalian cells. a**, Model system to interrogate gradual (ROS, blue) and acute mutation pressure (UV exposure, brown) on the mammalian genome. Triangles represent gradual accumulation of ROS mutations over several cell generations, whereas boxes represent fixed mutations in the genome after a specific cell division event. Schematic below depicts the experimental method from standard cell culture, exposure to UV, penning single cells, splitting two sister cells into separate pens after the first mitosis and finally proliferation/export. **b**, Images of single penned cells and FUCCI fluorophores imaged on the Phenomex Lightning platform (left), splitting mitotic sisters with light cages (middle), as well as the expanded populations (right). WGS, whole-genome sequencing.
**c**, A representative panel of cells ($n = 90$) imaged to ascertain the intensity of red and green fluorophores in 3-h intervals after penning. Cell cycle designation to the left of each cluster was determined by the fluorophore intensity at the time of penning seen in column 1 and denoted by the gray border. Subsequent heatmap columns represent 3 hour imaging timepoints of the same cell. Cell cycle color key circle is below. Models to the right implicate theoretical strand-specific (blue/yellow) distribution of DNA damage (red triangles) depending on the cell cycle phase at the time of UV treatment (timepoint 0).

Here, we present both cell culture and in vivo strategies to dissect mutagenic processes active in single cells. We adapted a microfluidics platform to track single mitotic events and analyze mutational patterns in mitotic sisters, revealing mirror-image and genome-wide mutation phasing. Our analyses revise the current model of transcription-associated ROS repair, demonstrate UV damage retention and confirm key predictions of the lesion segregation model[14]. Similarly, by exploiting a first generation mouse cross, we demonstrate that mutations can be phased to single-strands of DNA in liver tumors. The strategies we present can be broadly used to isolate and analyze the concurrent mutational processes driving transformation and cancer genome evolution.

## Results

### Separation of mammalian sister cells after a single mitosis

We exploited a microfluidics system to physically separate two mammalian sister cells following a single mitosis. This allowed us to test several hypotheses regarding DNA damage and repair. For example, established mutations in a single genome should be shared between mitotic sisters (Fig. 1a). In contrast, acute damage to the DNA of the parent cell should result in the damage, and resulting mutations, not being shared and segregating into separate daughter cells[14] unless the damage is resolved into a mutation before genome replication.

The Phenomex Lightning platform allows penning of individual cells and subsequent movement of these cells via light activation of a silicon membrane (Fig. 1b and Supplementary Video 1). Penning specifically refers to the process of moving a single cell into a single well, or pen, of the Phenomex chip. This platform can regularly image and physically separate sister cells after a single mitotic event. Subsequent whole-genome sequencing (WGS) of the expanded populations allowed us to determine the mutational landscape of these sister cells. Under optimized culture conditions (see Methods), cells divide at a rate comparable to that measured in standard cell culture (Extended Data Fig. 1a) and the genome typically remains diploid (Extended Data Fig. 1b).

To control for ploidy and cell cycle, we integrated the FastFUCCI construct[16] through lentiviral transduction into the nonadherent mouse cell line P388D1. FUCCI, or Fluorescent Ubiquitination-based Cell Cycle Indicator, uses a combination of fluorophores degraded at particular times in the cell cycle. The clone selected from this line (PF1) revealed fluorophore intensities correlated with DNA content and cell cycle phase (red cells: G1, green cells: G2/M; Fig. 1b,c and Extended Data Fig. 1c–h). The Phenomex system is equipped with lasers compatible with FUCCI fluorophores (Fig. 1b). Cell cycle progression of PF1 cells was observed as switches between a G2/M (green) to G1 (red) state (Extended Data Fig. 1c). In summary, we have established a controlled system to physically separate individual sister cells after a single mitotic cycle.

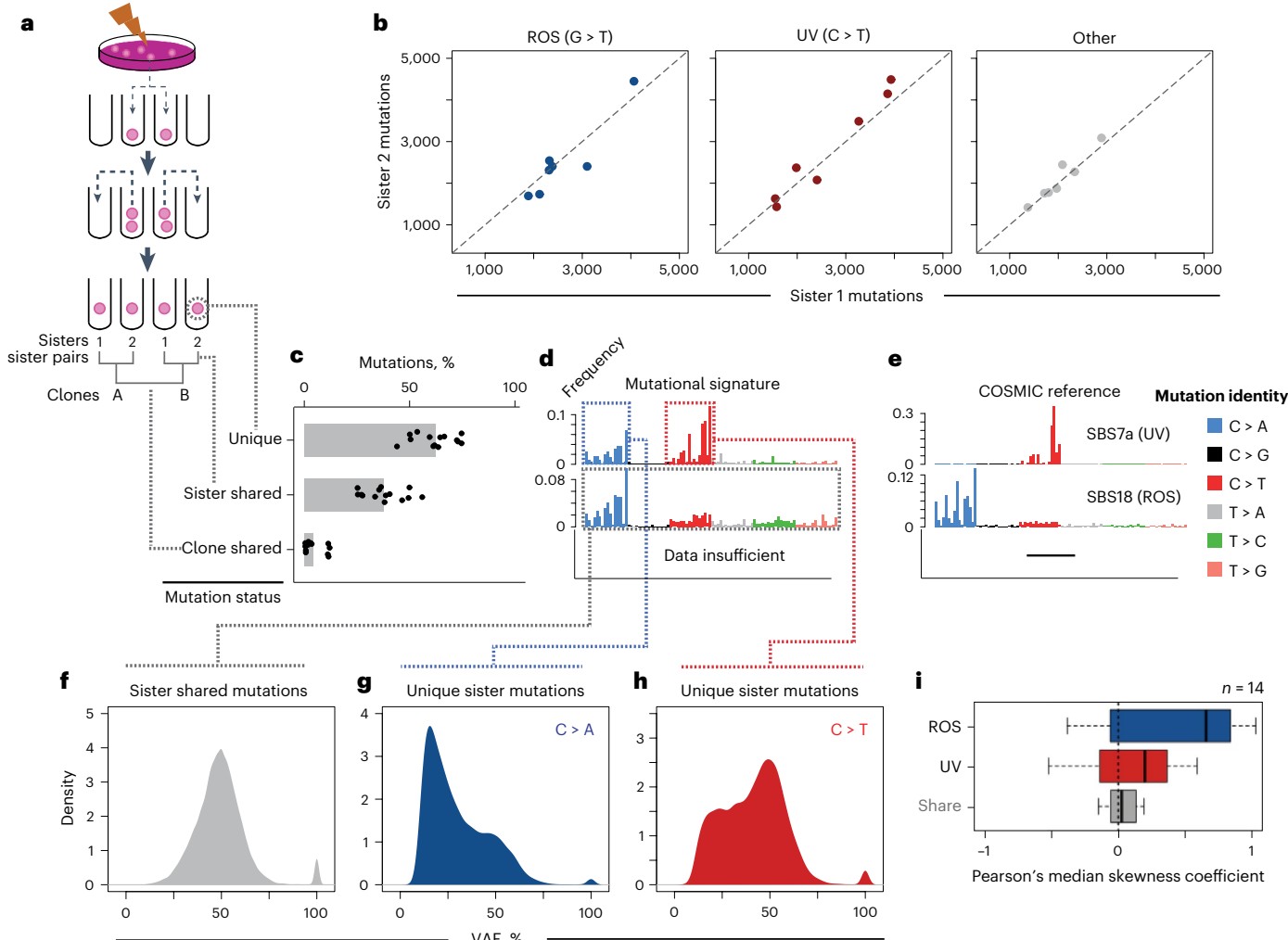

**Fig. 2 | Resolving acute and chronic mutational processes. a**, Schematic of separation experiment and nomenclature for mitotic sisters and clones. **b**, Number of mutations between sister pairs for distinct mutational processes; each point is one sister pair. Scatterplots represent the number of mutations between sisters for G > T (ROS, blue), C > T (UV, red) and other (gray). **c**, Barplot of mean overlap of mutations in percent between 7 mitotic sister pairs, consisting of 14 genomes. Unique refers to mutations for each single sister cell, sister shared represents mutations shared between mitotic sisters, and clone shared refers to mutations shared with other clones. Points are overlaid for each individual genome. **d**, Mutation signatures for each category in panel c, where each bar represents a specific type of mutation (identity key next to panel e) in a specific trinucleotide context (96 total bars). **e**, COSMIC reference signature with highest similarity to panel d. **f**, VAF for mutations shared between mitotic sisters. **g,h**, VAF as in panel f but for ROS mutations (g) and UV mutations (h) unique to each sister cell, respectively. **i**, Pearson's median skew of the VAF populations in panels f–h. Each box represents 14 measurements for the specific mutation category from each sister genome. Boxplot elements: median is the thick middle line, quartiles 1–3 are represented by the colored boxes, and whiskers denote the minimum and maximum of nonoutlier values.

## Distinguishing gradual and acute mutagenic processes

We reasoned that UV and ROS mutations were distinguishable for two reasons. First, both genotoxins have clear mutation signatures. ROS causes G > T transversions[17–19], whereas UV damage results in C > T transitions especially at dual pyrimidines[10,20]. Second, ROS exposure is chronic in cell culture, whereas exposure to UV treatment is a single damage event. This has direct implications for resulting variant allele frequencies (VAFs).

We treated PF1 cells with an acute 3-s dose of UVC such that approximately 50% of the population proliferated after exposure (Extended Data Fig. 2a). After UV treatment, we penned individual cells on the Phenomex platform and separated sisters after a single mitotic division. Hereafter, we refer to each unique single pinned cell as a clone, denoted by clone A, B, C, etc (Fig. 2a), and the terms 'sister 1' and 'sister 2' represent the first mitotic sisters. For example, clone A1 represents sister 1 from the singly penned clone A, whereas clone A2 represents sister 2 from the same singly penned clone A (Fig. 2a).

We expanded 14 daughters from 7 independent mitoses into clonal colonies large enough to perform WGS. Each genome was sequenced to ≥ 20x mean coverage, and mutations called against untreated cells. All clones contained ~6,000–9,000 mutations per genome (Fig. 2b) and mutation signatures had high similarity to COSMIC[20] SBS7a/b and SBS18, patterns attributed to UV[20,21] and ROS, respectively (Fig. 2e, Extended Data Fig. 2b, c). We reasoned that sister cells should contain a similar number of mutations arising from each mechanism. Indeed, mutation frequencies were nearly identical between mitotic sister genomes (Fig. 2b).

We surmised that mutations arising from acute UV damage should not be shared between mitotic sisters, because each sister inherits separate independently damaged strands. Roughly 90% of C > T transitions characteristic of UV damage were unique to a single mitotic sister (Fig. 2c,d, upper). In contrast, sister-shared mutations (~33% of total) were predominantly G > T transversions, sharing a signature most similar to SBS18[20] and suggesting they are ROS-induced mutations (Fig. 2d,

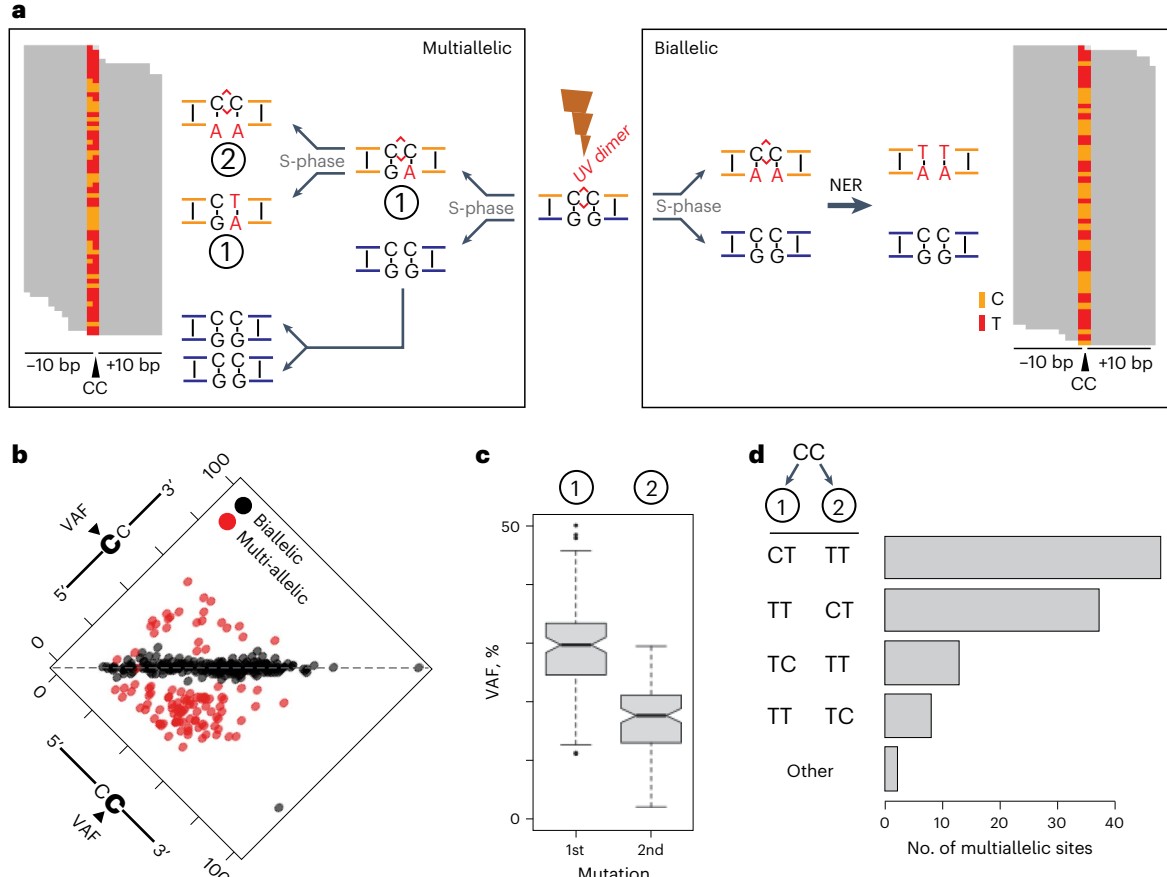

**Fig. 3 | Retained UV damage can drive multiallelic variation. a**, UV dimers (red lines/links, center) occur due to covalent linkages between consecutive bases along the DNA. Following S-phase replication, only one sister clone will inherit the lesion-containing DNA duplex (gold). Left: Lesion retention for two cell cycles can give rise to multiallelic variation, in this case evidence for CC > CT mutations (1), as well as CC > TT (2). Right: Schematic representation of a biallelic mutation occurring in one cell cycle, followed by NER of the lesion-containing strand. **b**, Scatterplot for VAF (%) of the second C in a CC dual mutation (*x* axis) and first C (*y* axis). Black points represent instances where only one alternative allele is detected, whereas red points contain at least two alternative alleles with at least three unique reads supporting the presence of each allele. The dotted line through the middle represents an identical VAF for both cytosines. **c**, VAF for the most common, and second most common allele at multiallelic sites (*n* = 111). The second allele has half the VAF of the most common variant, suggesting it was added one cell cycle later. Boxplot elements: Median is the thick middle line, quartiles 1–3 are represented by the gray box, 95% confidence interval is shown by notches, whiskers denote the minimum and maximum of nonoutlier values and outlier values are shown as points. **d**, Frequency of mutation types for the first mutation event (1) and second mutation event (2). 85% of the cases are CC > CT and CC > TT.

middle; and Extended Data Fig. 2c), which agrees with background mutations observed in cell culture[10] and 8-oxo-G mutagenesis in human dermal fibroblasts[22]. Finally, mutations shared between independent clones are rare (Fig. 2c), suggesting each clone is the result of a unique evolutionary trajectory.

We hypothesized that G > T transversions shared between mitotic sisters represent the landscape of ROS mutations present in the single clone when it was penned (Fig. 1a). If true, these mutations would have a VAF of 50%, as both sister cells inherit one mutated allele. In contrast, mutations accrued at later timepoints should have a positive skew in the VAF distribution. Indeed, shared mutations had a clear and distinct VAF of 50% (Fig. 2f; compare with Extended Data Fig. 2d). In contrast, sister-unique G > T transversions had a reduced VAF and more positive skewing in the VAF distribution (Fig. 2i), suggesting they arise after sister clone separation (Fig. 2g). The hypothesis that shared mutations were ancestral is supported by the observation that the four sister genomes with higher clonal mutation sharing (Fig. 2c, bottom) are derived from two singly penned clones on the same Phenomex chip (Extended Data Fig. 2e). Finally removal of shared mutations increased similarity to SBS7a (Extended Data Fig. 2f). In conclusion, UV mutations are unique to each sister while

ROS mutations can be subdivided into ancestral and sister-specific accrued mutations.

## UV lesion retention causes multiallelic variation

The bimodal VAF distribution for UV mutations suggested rounds of nonmutagenic replication over persistent lesions (Fig. 2h). Such lesion persistence could allow for the incorporation of distinct alternate bases from consecutive S-phases, a phenomenon termed multiallelism[14]. Multiallelism is the observation of more than one alternative allele at a single genomic position. We sought to determine if multiallelism could be observed for mutations with a UV signature in our data.

Identifying multiallelism for UV damage is challenging because only (C > T) transitions are observed at a single position. We therefore extended our observation to tandem CC mutations, because three possible alleles can result (CC > TT, CC > CT and CC > TC). Although dinucleotide mutations represent only ~2.2% of all UV mutations across the 7 sister pairs sequenced, the well-described[23,24] CC > TT event (Extended Data Fig. 3a–c) is most common[23,24]. We identified reads fully overlapping CC > TT tandem mutation events (*n* = 373 sites), and discovered the presence of more than one alternative allele in single clones supported by read-level data (Fig. 3a, left). This indicated that an unrepaired UV lesion

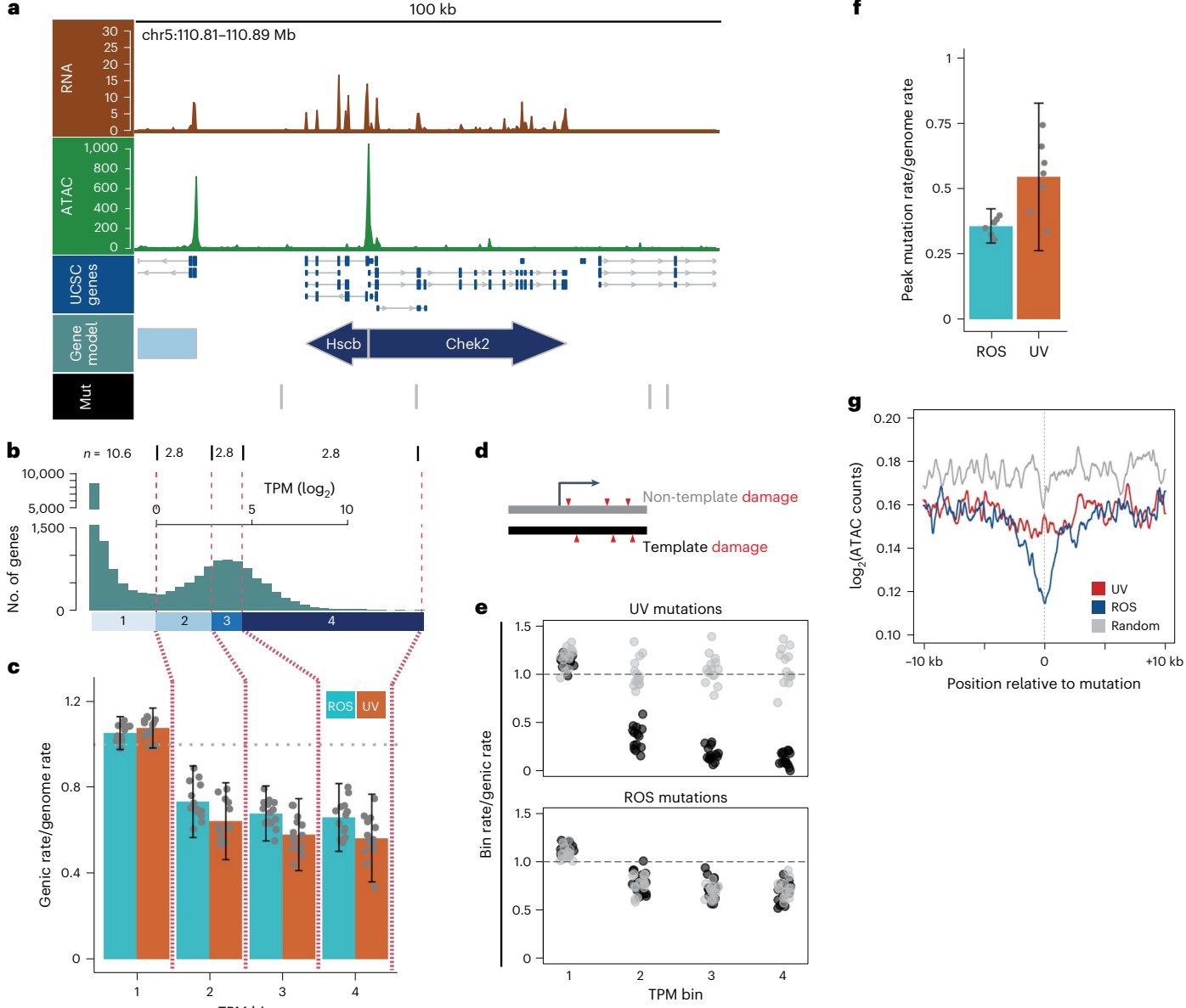

**Fig. 4 | Differential repair for UV and ROS in transcribed regions. a**, Genomic screenshot of a 100 kb region of chromosome 5, from top to bottom showing RNA-seq reads (brown, coverage), ATAC-seq reads (green, coverage), UCSC gene annotation (blue), gene model based on ATAC and RNA data (light blue) and mutations (Mut) in the region (black vertical lines, bottom track). Gene arrows are colored based on their expression bin as noted in boxes under the histogram in panel b. **b**, Histogram of $\log_2(\text{TPM})$ measurements for all mouse RefSeq genes at least 1 kb from the nearest gene and 1 kb in length (19,091 total). Binning based on quantiles of TPM measurements are shown as vertical red dashed lines. Numbers above the dashes represent the number of genes (in thousands) in the respective bin. **c**, Mutation rates in gene bodies for all genes in bins outlined in panel b. *y* axis represents the observed genetic mutation rate for a bin divided by the expected mutation rate (calculated using all mutations). Horizontal gray dashed line represents genome average. Error bars represent 2 standard deviations from the mean of 14 genomes. **d**, Cartoon depicting stranded damage, which can be interrogated for gene bodies. **e**, Mutation rates for UV (upper) and ROS (lower), separated into template (black) and non-template (gray) mutation rates. Each point represents one sister genome, and bins are as shown in panels b and c. *y* axis is the mutation rate in the genetic bin divided by the total genetic mutation rate. The horizontal gray line represents the expected rate based on all genomic mutations. **f**, Observed mutation rates within ATAC peaks (accessible regions) divided by the genome mutation rate for each source. Mutations for sister pairs were combined to calculate delta rates and bars depict the mean across all seven combined genomes. Individual points for each genome are overlaid. **g**, Accessibility metaplot around UV (red) and ROS (blue) mutations. Accessibility is shown as the log2 converted average number of reads at positions flanking the mutation. A 201-bp smoothing window was applied to these averages.

can result in the generation of both dinucleotide and mononucleotide substitutions at the same site. This observation is distinct from biallelic mutations previously reported in melanoma where both haplotypes are mutated[25]. We confirmed and quantified this observation in a genome alignment independent manner (Methods and Extended Data Fig. 3d–g).

We next asked whether both bases at each CC > TT dual mutation site have the same VAF. Differing VAFs between neighboring

cytosines would provide evidence that each mutation was fixed in a different cell cycle (Fig. 3a, left). In contrast, identical VAFs between neighboring cytosines indicates a simultaneous mutational event (Fig. 3a, right). Identical VAFs (biallelic mutations) were observed for 70.2% of CC > TT dual mutations (Fig. 3b, black points; n = 262 sites), while 29.8% of VAF pairs at tandem mutation sites were different (Fig. 3b, red points). This analysis revealed

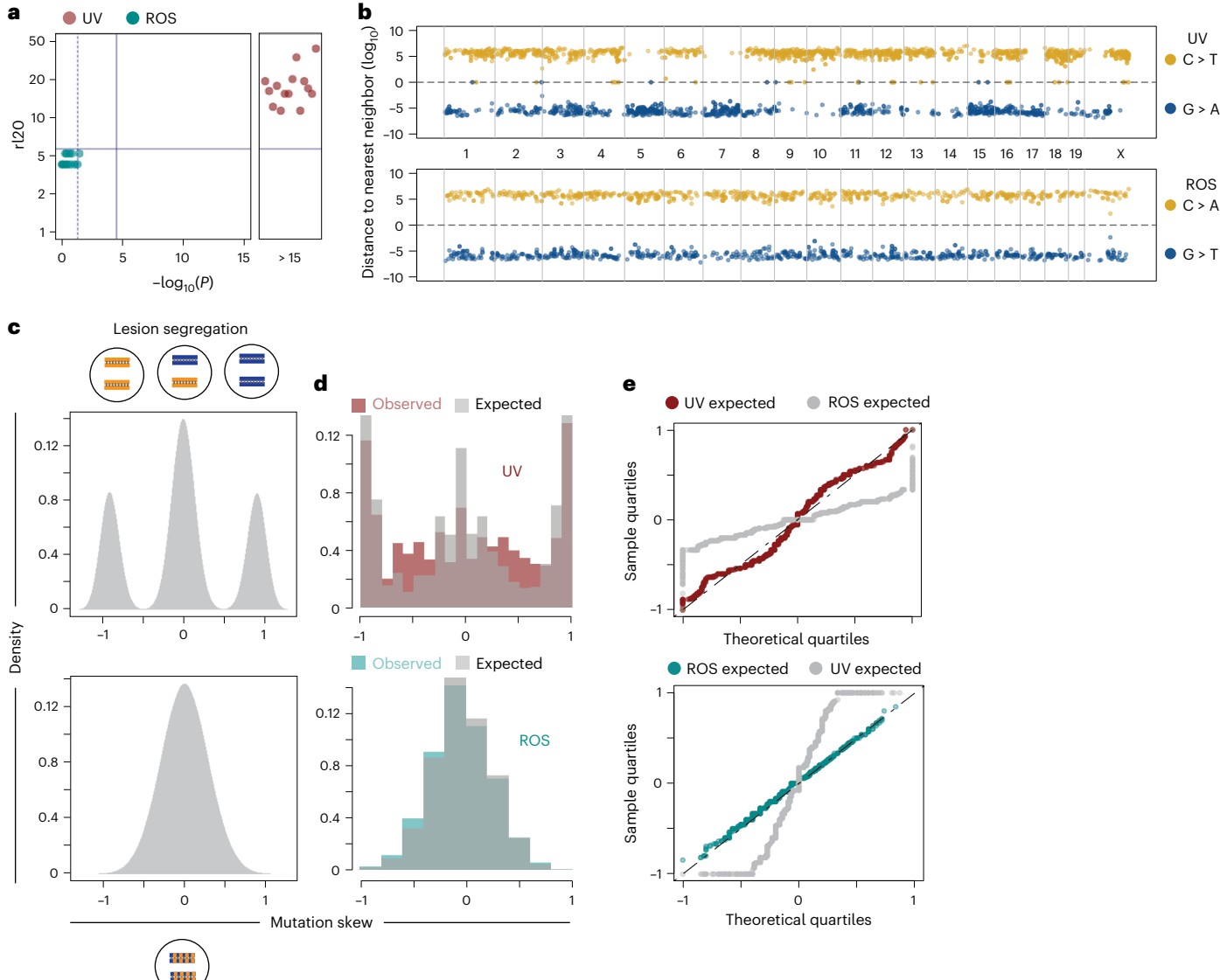

**Fig. 5 | Mutational phasing via acute UV damage is established in a single cell cycle. a**, rl20 metric (see Methods) used to determine if there is significant evidence for runs of a single mutation type. The y axis depicts the longest set of runs for a single mutation type, accounting for 20% of all informative mutations. The x axis is the significance (p-value) of seeing such a run length given random assignment to strands, calculated using a two-sided Wald–Wolfowitz runs test. Light blue points represent C > A or G > T mutations (ROS) analyzed for each genome. Red points represent C > T or G > A mutations (UV). **b**, Example of lesion segregation pattern due to UV induced mutations (top) across all chromosomes of a single mitotic sister, and lack of phasing from ROS induced mutagenesis (bottom) in the same sister. Reference cytosine mutations are shown as yellow dots, whereas reference guanine residues are in blue. The y axis represents log2(distance to nearest neighbor), with G mutation distances converted to negative values to distinguish them from C residues. Chromosomal boundaries are denoted by black vertical dashed lines and chromosomes noted between the tracks. Horizontal dashed line represents a distance of 0. **c**, Theoretical distribution of mutation phasing in lesion segregation (upper) and no phasing (lower) in 10-Mb genomic tiles. **d**, Sampling of the distributions of C using the same number of tiles actually profiled in the data (gray), and distribution of the skew from 10-Mb genomic tiles for all 14 genomes profiling acute UV (upper, red) and chronic ROS (lower, light blue) mutagenesis. **e**, QQ plot comparing distributions of ROS, UV and random sampling of the respective models in teal, red and gray respectively.

that multiple alternative alleles were present at a subset of tandem mutation sites.

We next reasoned that VAFs at multiallelic sites could reveal the order by which mutations were introduced. Indeed, the observed 2:1 ratio of the VAFs for the first and second most common alternate alleles was consistent with multiple error-prone replication events over an unrepaired pyrimidine dimer (Fig. 3c). The dominant sequence mutated within these dimers is the 3′ cytosine, occurring in 85 of 111 multiallelic sites (Fig. 3d and Extended Data Fig. 3c,d). Situations where an initial 3′ cytosine mutation was followed by a double cytosine mutation in the next mitosis were approximately as common as the opposite order. In contrast, alleles with a single mutation at the 5′ cytosine occur

in 22.2% of multiallelic sites. Almost no combination of single mutation events at these loci was observed, such as CC > CT followed by CC > TC. We considered that either an adjacent pyrimidine, or multiple adjacent pyrimidine dimers, located 3′ to the CC site could account for the observed 3′ bias. To test this, we measured base composition surrounding multiallelic sites and compared it to both biallelic and randomly sampled CC sites across the genome. On the contrary, pyrimidines were significantly enriched directly 5′ from CC dual mutations ($p < 0.005$, Fisher's exact test, Benjamini-Hochberg corrected, Extended Data Fig. 3h), arguing that the 3′ bias is not caused by a dimer directly downstream of multiallelic sites. Nevertheless, we cannot completely rule out that an additional dimer in an NpCpCpN context influences the

mutation bias. Taken together, we reveal UV-induced multiallelic variation at dual CC sites, and that most consecutive mutation events are CC > CT and CC > TT, in either order.

## Transcription-associated ROS repair is strand agnostic

We next asked how UV and ROS mutation rates are affected by transcriptional activity and chromatin accessibility. TCR is a well-documented phenomenon[26] from prokaryotes[27] to humans[28], where DNA damage causes a transcribing RNA polymerase to stall, triggering NER[29]. We first determined bulk RNA levels from the PF1 line (Fig. 4a and Extended Data Fig. 4a,b) and binned genes based on tags per million (TPM) (Fig. 4b) (bin 1 = 10,6,000 unexpressed genes, bins 2–4 = ~2,800 expressed genes). As expected, active transcription significantly suppressed mutational rates ascribed to both UV[28–31] and ROS[32–36] ($P = 1.7 \times 10^{-6}$, two-tailed Mann-Whitney test; Fig. 4c). Both mutation rates were lowest in highly transcribed gene bodies, consistent with previous estimates for TCR in the case of UV (Fig. 4c)[31].

A hallmark of TCR is that lesions are repaired specifically on the template strand[37,38]. Given the well-described UV mutation mechanism, C > T mutations in minus-strand genes or G > A mutations overlapping plus-strand genes result from damage to the template strand (Fig. 4d). As expected, the template strand mutation rate in expressed genes is approximately 10% of the total genomic rate of UV-associated mutagenesis (Fig. 4e, upper), which is likely an underestimate of the repair rate (Extended Data Fig. 5). Using the same rationale, we assigned ROS mutations to either the template or non-template strand assuming 8-oxo-G as the lesion[22]. Contrasting with prior observations[39], ROS repair appeared to occur equally on both template and non-template strands (Fig. 4e, lower) for both shared and unique ROS mutations (Extended Data Fig. 4f). Together, these data demonstrate that transcriptional activity correlates with reduced mutation rate from both UV and ROS, yet only UV repair is strand asymmetric.

The reduction in ROS mutation rates within transcribed gene bodies was expected[32,34,35], but the symmetry in template and non-template mutation rates was unanticipated[40]. We asked whether local chromatin accessibility as measured by ATAC-seq (Extended Data Fig. 4c–e) may be partially responsible. Globally, mutation rates within ATAC peak regions for UV and ROS were 50% and 32% of the genome average, respectively (Fig. 4f), in agreement with higher repair rates in open chromatin[40]. There was a clear decrease in chromatin accessibility around both UV (37,756) and ROS (28,184) mutations (Fig. 4g). Furthermore, ROS mutations revealed a stronger local depletion in ATAC signal, suggesting that BER is less efficient in detecting damaged bases in more highly compacted chromatin. Actively transcribed gene bodies tend to have greater accessibility than silent genes (Extended Data Fig. 4g) as well as increased OGG1 recruitment (Extended Data Fig. 4h, previously published ChIP data[41]). In sum, our data suggest that the decrease in ROS mutation rates in transcribed gene bodies is not triggered by canonical TCR detection but at least in part due to increased local accessibility for repair machinery.

## Mutational phasing is a unique to acute mutagenic exposure

A prediction of the lesion segregation model[14] is that a single, acute mutagenic exposure will cause mutation strand phasing. In contrast,

although ROS damage is likely subjected to lesion segregation during each cell division, its gradual accumulation over many cell generations would be expected to erase any mutational asymmetry. Our system provides an ideal landscape to test this hypothesis, where acute and chronic mutational processes occur in the same genome yet their signatures can be readily identified.

We separately considered the mutation counts and genomic distribution of SBS7 UV mutations (C > T / G > A) and SBS18 ROS mutations (C > A / G > T) (Extended Data Fig. 5a), which confirmed the expectation of chromosome-scale mutational asymmetry and strand-phased UV mutations in sister clones (Fig. 5a,b; $P < 1 \times 10^{-15}$; permutation based rl20 metric[14]). In contrast, ROS mutations did not show mutational asymmetry in bulk analysis (Fig. 5a,b) or when partitioned into sister-shared and sister-unique mutations (Extended Data Fig. 5b). Direct comparison to Bernoulli models (Fig. 5c,d and Extended Data Fig. 5c,d) confirms the strand distribution of UV mutations is a good fit to a single burst of DNA damage followed by lesion segregation with random strand retention (Fig. 5e, upper). In contrast, the strand distribution of ROS mutations closely matches expectation for the random assignment of mutations to DNA strands (Fig. 5e, lower). Taken together, these results suggest pulse UV mutagenesis is fixed during post-exposure mitosis in a strand-specific manner, whereas chronic ROS exposure resembles progressive mutation accumulation over many cell generations.

## Mirror-image mutation phasing in mitotic sisters

In the first mitosis following UV damage, complementary lesion-containing DNA strands are expected to segregate into separate daughter cells (Fig. 6a). When both copies of a diploid chromosome inherit the C > T phase of lesions in mitotic sister 1, mitotic sister 2 exhibited the complementary G > A asymmetry (Fig. 6b, chromosome 17). When mitotic sister 1 inherits a C > T and a G > A allele, the same occurred in mitotic sister 2, resulting in a mix of both mutation types (Fig. 6b, chromosome 8). If a sister chromatid exchange (SCE) event occurred[14], equivalent positions on the affected chromosome showed a transition from mixed to opposite asymmetry between mitotic sisters (Fig. 6b, chromosome 2). At the genome scale, sister cells shared mixed segments while revealing opposite asymmetry in phased regions (Fig. 6c). Notably, all seven pairs of mitotic sisters exhibited mirror-image mutational asymmetry across their genomes (Fig. 6d).

We sought to quantify the significance of this mirror-image mutation phenotype by comparing mutation skew in 10-Mb genomic windows for all clones and sister pairs. For mitotic sisters, mirror-image phasing patterns should result in a linear negative relationship. For unrelated clones, the prediction would be a random relationship in their phasing because their UV mutations are independent damage events. In our data, on average, segment phasing patterns of sister 1 explained 49% of the variance in phasing patterns for sister 2 from a single division event. In contrast, the variance in phasing patterns for clone A explained less than 1% of the variance in phasing for clone B (Fig. 6e). In sum, these findings demonstrate that acute UV damage produces mirror-image mutation phasing patterns in mitotic sisters, independent of selection.

**Fig. 6 | Mitotic sisters have mirror-image mutation distributions across the genome. a**, Model of lesion segregation creating mirror-image mutation patterns between mitotic sisters. Replication of damaged DNA creates stranded mutations, and sister cells from the subsequent division inherit either only one type of strand (upper) or one of each strand (middle) or can undergo SCE and switch from a mixed to a phased segment (lower). **b**, Representative examples for each strand inheritance type depicted in panel a. **c**, Scatterplot of C (yellow) and G (blue) UV mutations across all chromosomes for one mitotic sister pair. Lightly colored yellow/blue/gray background represents segmentation from changepoint analysis. **d**, Segmentation heatmap for all 7 pairs (14 total genomes). Color legend of phasing noted below heatmap, mitotic sisters are adjacent to

each other with a white gap between clones. **e**, Test for mirror-image mutation patterns. Heatmap represents the correlation coefficient (Pearson's $R$) for mutation skew between mitotic sisters and clones. Smooth scatterplots to the right represent skew correlations between an example of mitotic sisters (top) and clones (bottom). **f**, Diagram demonstrating how recombined strand mutations are determined. **g**, Metaplot of mutations/megabase (mu/Mb) for recombined and non-recombinant strands (brown and gold respectively), as well as mutation density for regions where one strand is not uniquely assignable. Window represents 20-Mb flanking the SCE site shown at 0 (vertical gray line). Individual points represent smoothed 1-Mb sliding windows with a 100-kb step size.

We next explored another specific segmentation phenotype: switches in phasing within a single chromosome (Fig. 6a,b, bottom panels). If these switches reflect sister chromatid exchange, then reciprocity between two mitotic sisters should be observed (Fig. 6a, bottom). Indeed, this segment switching was evident at 130 total positions across all samples, with approximately one crossover event per

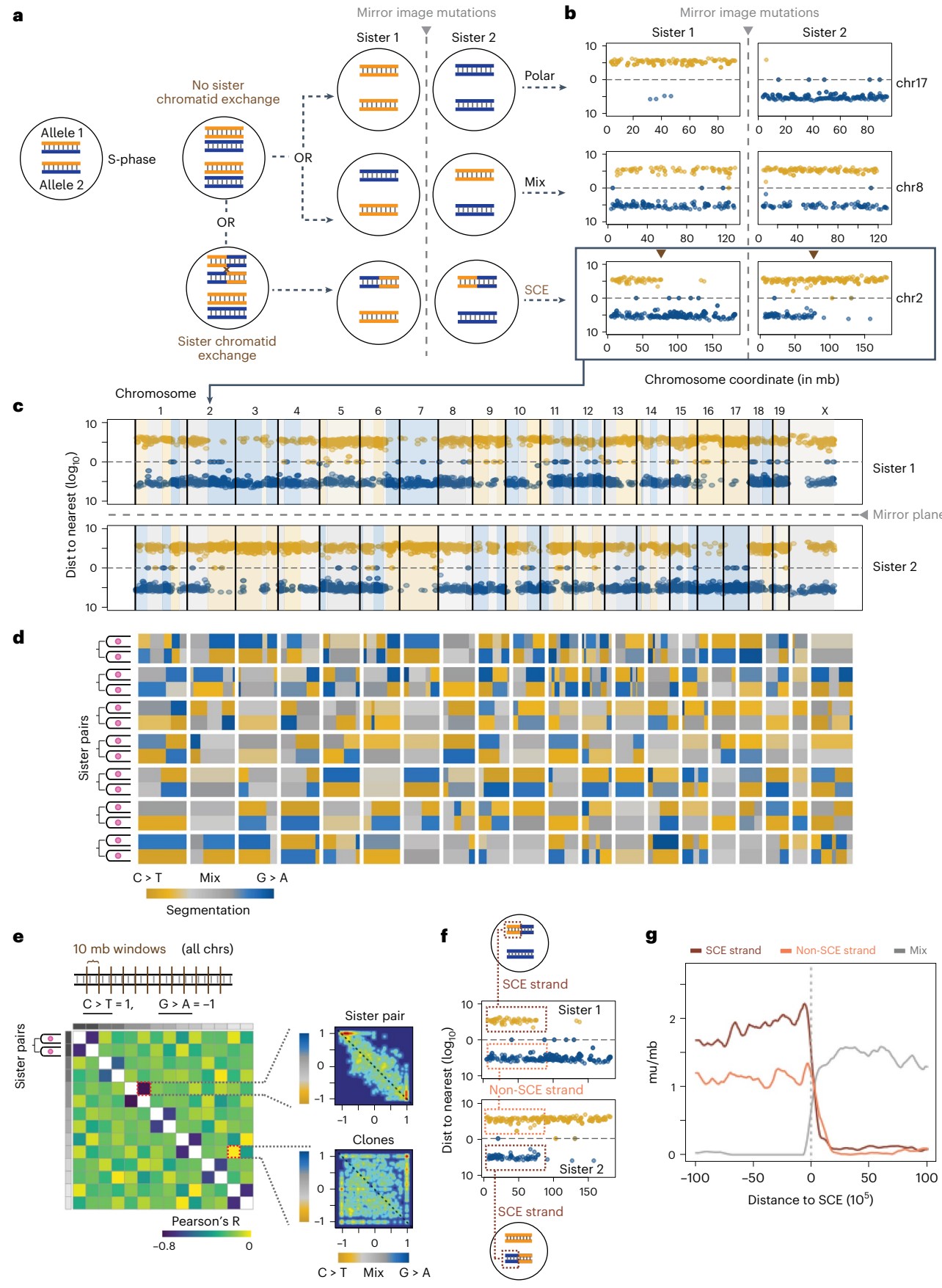

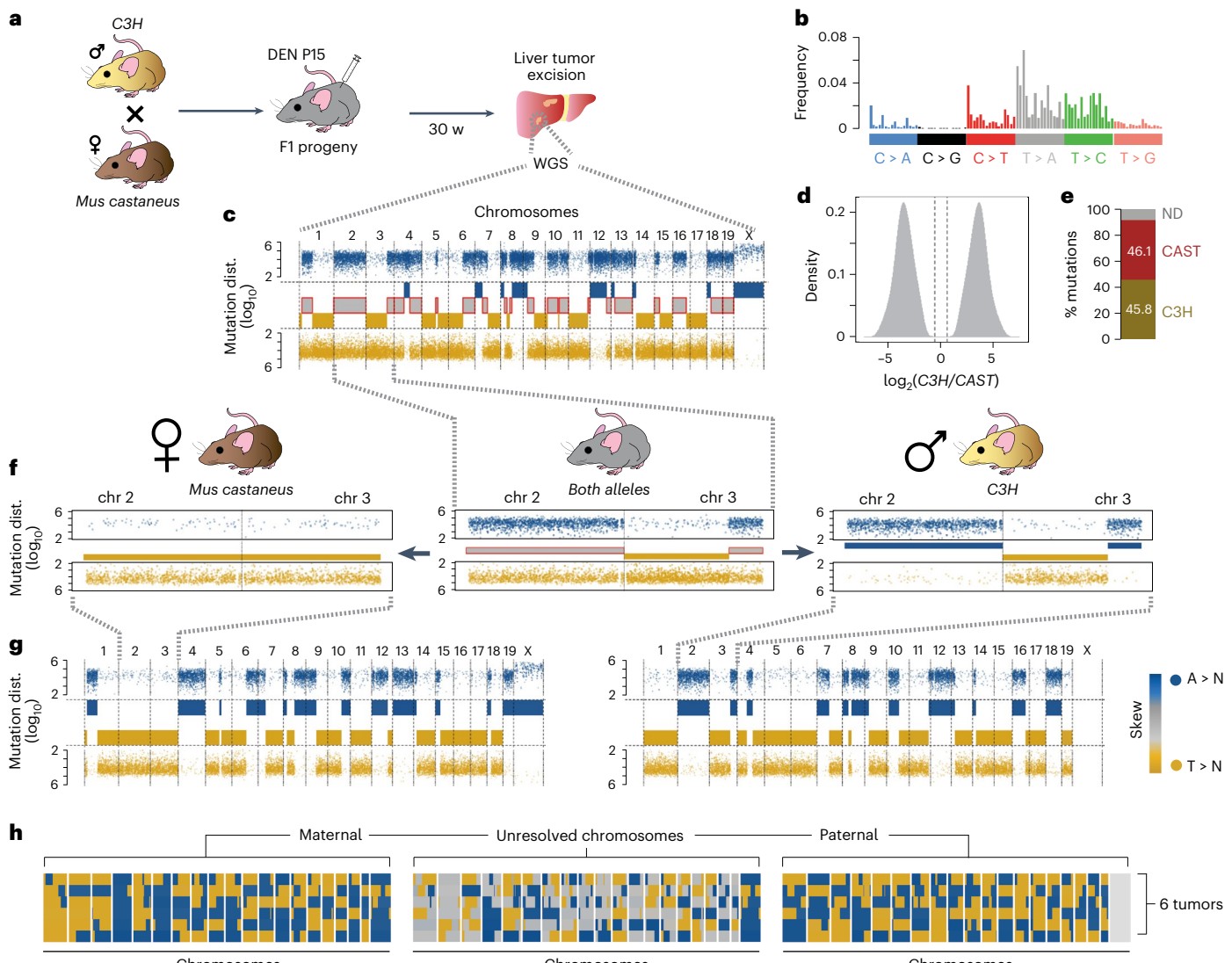

**Fig. 7 | Allelic resolution of lesion segregation *in vivo*. a**, Experimental design: a *Mus musculus castaneus* female was crossed with a C3H male. Two 15-day-old (P15) progeny were injected with DEN and 3 tumors from each mouse were isolated 30 weeks later for WGS. **b**, Mutation signature of tumor mutations. **c**, Lesion segregation plot of a representative tumor with haplotype-agnostic data. Segmentation boxes in the middle reflect if the chromosomal segment has mixed (gray with red outline) or phased mutations. Reference A mutations are shown in blue, and reference T mutations are shown in gold. **d**, Frequency distribution of species specific delta read counts for all mutations in the sequenced population with at least 2 unique haplotype assignable reads. Gray vertical lines denote the cutoff for species specificity. **e**, Proportion of all mutations assigned to the C3H

haplotype (C3H, gold), *M. castaneus* (CAST, brown) or undetermined (ND, gray). Actual percentages for haplotype-specific reads are shown in the respective bars. **f**, Chromosomes 2 and 3 from the tumor depicted in panel c. Middle panel shows mutation distribution (points) and segmentation (bars in the middle) for haplotype-agnostic data. *M. castaneus* (left) and *C3H* (right) show the same plots but after resolving haplotype. Note that the mixed segments become completely resolved into phased mutation stretches. **g**, Plotting the mutations after assignment to one haplotype. **h**, Segmentation before (middle) and after (left and right) haplotype resolution of mutations. Heatmap represents tumor segmentation phasing for the tumor in panel c and five additional tumors (six in total).

chromosome (mean 0.92; Fig. 6b–d). As both normal lymphoblast and fibroblast cells from healthy human patients are estimated to have roughly five SCE events per mitosis[42], our fourfold elevation suggests UV damage is responsible for this increase in SCE, consistent with previous studies[43–45]. There was a modest increase in mutation density around SCE sites (Extended Data Fig. 6c), suggesting locally increased UV damage may trigger an SCE event. We separated mutations based on whether they originated from the recombined or non-recombined strand (Fig. 6f), revealing a clear trend for increased mutation density on the strand undergoing recombination (Fig. 6g). In conclusion, elevated local UV damage rates on one strand of DNA are more likely to trigger an SCE event during the first mitotic cycle.

## Haplotype resolved mutations in F1 mice liver tumors

In *N*-nitrosodiethylamine (DEN)-induced liver tumors from inbred male mice, the vast majority (~95%) of mutations are phased on the hemizygous X chromosome[14]. We reasoned that assigning mutations to a single haplotype would reveal a similar phasing phenotype across the entire genome. An ideal model for this is the F1 progeny from a mouse cross where the parents have substantial germline single-nucleotide polymorphisms (SNPs). The C3H and *Mus castaneus* (CAST) mouse subspecies produce viable offspring and their genomes differ by 20 million SNPs[46]. Given the SNP genomic distribution, paired-end 100-bp sequencing allows ~60% on average of all unique reads to be assigned to a single haplotype (Extended Data Fig. 7a,b and Methods).

We injected two F1 mice at P15 from a CAST and C3H cross with DEN and isolated 6 tumors 30 weeks later (Fig. 7a and Methods). We carried out WGS (≥20× mean; Extended Data Fig. 7c) of these tumors and used N-masking[47] to map reads to a modified C57BL/6 reference genome. Acute DEN treatment predominantly induced mutations by damaging T bases (38.5% T, 37.3% A, 12.4% C, 11.8% G) with total mutations, mutational signature, and stable chromosome copy number as documented previously[14] (Fig. 7b and Extended Data Fig. 7d,e). Haplotype-agnostic mutations reproduced a lesion segregation phenotype identical to that found in the inbred parental strains (Fig. 7c).

By exploiting the co-occurrence of germline SNPs with mutations on the same read, we could phase 91.9% of mutations to either the C3H or CAST genomes (Fig. 7d,e and Extended Data Fig. 7d). We used the single copy X chromosome to estimate our false assignment rate as ~0.27% (s.d. = 0.25). Once mutations were accurately assigned by haplotype, segments with mixed mutational patterns resolved into two phased component alleles (Fig. 7f). For example, in the absence of haplotype assignment, Chromosome 2 of a representative tumor is a mixture of T > N and A > N mutations. Resolving the mutations by haplotype reveals that the CAST allele contains the vast majority of T > N mutations, whereas the C3H allele has predominantly A > N mutations. Switches in mutational phasing are now clearly resolved to a single, recombined allele (Fig. 7f, chr3).

After segmenting haplotype resolved mutations using changepoint[48] analysis, the mutational landscape of each autosome was virtually identical to that found on the single X chromosome (Fig. 7g). In total, 95.6% (standard deviation = 0.92%) of allelic mutations are in agreement with their phase, comparable to the X chromosome (94.9%; Extended Data Fig. 7f). This relationship held true for five additional tumors from these two mice (Fig. 7h). Taken together, resolving mutations to single alleles demonstrates complete mutational phasing across the genome.

We next tested whether the T < > A mutation phasing pattern found in each tumor genome was independent of the phasing found in other tumors. In agreement with this assumption, phasing in 10-Mb bins revealed an average Pearson's correlation of 0.07 when compared between tumors (Extended Data Fig. 7g right, h-I). Although C > N and G > N mutations make up only ~20% of the data, these mutations are also phased in a manner analogous to T < > A segments (Extended Data Fig. 7g, left). Given T lesions result in the predominant DEN mutation signature, this suggests that damaged pyrimidine bases cause the majority of tumor mutations. Furthermore, C < > G phasing recapitulates intrachromosomal T < > A phase switches (140/140), suggesting these are indeed sister chromatid recombination events. Taken together, the use of F1 mouse crosses allowed DEN-induced mutational phasing to be assigned to single DNA strands.

## Discussion

Cancer genomes chronically accumulate genetic changes in parallel among competing clones[49–51]. Here, we describe a method to identify and disentangle two distinct genomic mutation processes in single cells: acute UV and chronic ROS. We interrogate mutations in mammalian mitotic sister cell populations post UV exposure. Our data show that UV lesions can persist for more than one cell cycle, driving multiallelic variation in CC dinucleotides. Mitotic sisters inherit genomic mirror-image mutation phasing, following a random inheritance of strands, analogous to a set of Bernoulli trials. Reciprocal intrachromosomal switches in phasing provide direct evidence of sister chromatid exchanges. In contrast, chronic ROS damage does not show strand-specificity in phased mutations or transcription-associated repair. Lastly, we demonstrate at a single chromatid level that almost all mutations are phased in tumors from F1 mice.

Our repurposing of a microfluidics system uniquely accounts for cell cycle state, cell division number and expansion rate. Analyzed cell lines can be genetically modified and/or be tested with a diverse array of mutagens. Similarly, F1 animals from inbred subspecies minimize genetic heterogeneity while providing haplotype-specific information. Our model systems have limitations. First, Phenomex requires nonadherent cells. Second, our strategies do not perfectly recapitulate specific aspects of human tumors, including cell cycle, genetic heterogeneity, and mutagenic exposure. Our mechanistic insights are nevertheless relevant to human cancers because: (1) phased mutational profiles are seen in patient data[14,52], (2) UV mutation signatures are found in human skin cancers, and (3) we recapitulate reciprocal mutation phasing of the mitotic sister predicted to occur at transformation but lost during clonal expansion. In addition, we show that consistent with recent findings from human development and homeostasis[53], multiallelic variation is observable in the absence of transformation. These findings provide further evidence that lesion segregation is a ubiquitous feature of mutagenesis following DNA damage.

Cancer genome evolution is considered a Darwinian process[54] wherein random mutagenesis and selective pressure determine clonal expansion. Our approach exploits microfluidics to separate sister cells and decouple mutational pressure from positive selection, enabling the fate tracking of the forward and reverse strands of both alleles. We establish in our system that UV damage is resolved into phased mutations following DNA synthesis. This mutation pattern parallels a model where clonal expansion of a primordial cell creates shared ancestral mutations[55]. In our case, sister cell populations continue to accumulate ROS damage, resulting in a heterogeneous mixture of mutations in punctuated equilibrium[56]. Cancer genome analysis could exploit mutational phasing as a fingerprint of the originally transformed cell.

How oxidatively damaged bases such as 8-oxoG[8] are repaired remains actively debated[57–59]. To date, nucleosomal positioning[60], transcriptional activity[36,61] and the FACT complex[62] have been implicated in resolving ROS damage, and our results newly inform how chromatin accessibility may also contribute. First, we can assign base damage to a specific DNA strand, because the dominant C > A mutation likely arises from oxidized G bases. Second, our method can analyze how UV and ROS damage are simultaneously handled by parallel repair processes in the same cell. Third, we found that pyrimidine dimers do not alter the repair of ROS lesions associated with transcription, because we can distinguish oxidative damage occurring before and after UV exposure. Our data collectively support a model where oxidative damage is repaired better in transcribed gene bodies but strand agnostically. Although stranded BER has been described in yeast[34] and humans[39] and for other oxidized bases[35], in our system, it does not appear to shape the repair of 8-oxo-G.

Our work decouples mutagenesis from selection to study how DNA damage shapes the mammalian genome. This framework for mechanistic analysis can be flexibly applied to separate otherwise-confounding mutational processes that co-occur in cells.

## Online content

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

## Methods

### Tumor induction in CASTxC3H F1 mice

Animal experimentation was carried out in accordance with the Animals (Scientific Procedures) Act 1986 (United Kingdom) and with the approval of the Cancer Research UK Cambridge Institute Animal Welfare and Ethical Review Body. Inbred female *M. musculus castaneus* (CAST/EiJ) mice were crossed with inbred male C3H/HeOuJ (C3H) mice. The F1 male offspring were treated with a single intraperitoneal dose of DEN (Sigma-Aldrich N0258; 20 mg kg$^{-1}$ body weight) as described previously[14]. Liver tumors were isolated 30 weeks after treatment, flash frozen in liquid nitrogen and stored at −80 °C for DNA extraction and sequencing. Liver tissue from an untreated P15 litter mate was sampled for control experiments. Control samples (liver tissue) were also collected from untreated, age-matched littermates. Animals were maintained using standard husbandry: mice were group housed at room temperature/humidity in Tecniplast GM500 IVC cages with a 12 h:12 h light:dark cycle and ad libitum access to water, food (LabDiet 5058), and environmental enrichments.

### F1 liver tumor genomic DNA extraction

Liver DNA was extracted using the Qiagen AllPrep DNA/RNA Mini Kit. Approximately 30 mg tumor was placed in 600 µl buffer RLT supplemented with 10 µl ß-mercaptoethanol. A 5 mM stainless steel bead (Qiagen, #69989) was added and the sample shaken for 2 x 20 s at 15 Hz on the Qiagen TissueLyser II. The lysate was centrifuged through an AllPrep DNA spin column for 30 s at 8,000 × $g$, washed with 500 µl buffer AW1 and 500 µl AW2 and eluted in 100 µl buffer AE.

### Cell culture and splitting on the Phenomex Lightning platform

P388D1 cells from ATCC were cultured before and during incubation on the Phenomex Lightning system using 5% C0$_2$ at 37 °C in DMEM supplemented with 10% fetal bovine serum, 2.5 mM L-glutamine, 1x Pen-Strep and 5x B27. The purpose of excess B27 was to reduce free radicals in the media. The incubator proliferation assay was carried out in triplicate by plating 2.5 × 10$^5$ cells in six-well plates. At 24-h timepoints, live cells were counted using trypan blue and a Countess 3. Doubling rates for incubator cells were calculated as for Phenomex (below). For Phenomex cell cycle FUCCI measurements (Extended Data Fig. 1c), cells were imaged in 3-h windows over the course of 15 h.

Before Phenomex penning, a full clean was carried out as per the manufacturer's instructions. After UV treatment (<30 min), PF1 cells at a concentration of 2 × 10$^6$/mL were penned as singlets and images acquired in ambient light, FITC and Texas Red excitation, and cell number/pen counted using the Jurkat CNN algorithm. One day after penning, cells were assayed for doublings using the cell analysis suite of Phenomex and singlet cells that were in G1 at time of penning (Texas Red, no FITC) were split. Default settings for manual OEP were used to move cells except wavelength voltage was increased to 6. Cells proliferated for 4–6 days and were exported to 96-well plates. Approximately 2 × 10$^6$ clonal cells were used in DNA extraction and WGS library preparation.

For calculating doubling rates, cell numbers were obtained over a 64-h period using a minimum of four measurements. Pens that did not proliferate were removed, and a linear model was fit whereby the log2(cell counts) were regressed on time in hours of culture. The mean adjusted $R^2$ for all fits was 0.905 with a standard deviation of 0.166 ($n$ = 998). The slope of the fit represented doublings per hour and was multiplied by 24 to represent divisions per day.

### Introduction of the FastFUCCI system in P388D1

The lentiviral vector pBOB-EF1-FastFUCCI-Puro obtained by AddGene was transformed into DH5α E.coli and midi-prepped (Qiagen MidiKit). Lentiviral packaging vectors VSVG and R8.91 were a kind gift from the lab of Michaela Frye. 5 × 10$^6$ Lentix HEK293T cells from Takara were transfected with 12, 5 and 12 µg FastFUCCI, VSVG and R8.91 vectors using Lipofectamine 3000. Media from day 2 and 3 of the transfected Lentix cells was sterile filtered (0.45 µM) and ultracentrifuged @ 77,125 × $g$ @ 4 °C for 90 min. The pelleted virus was resuspended in 100 µl Opti-mem media. 1 × 10$^6$ P388D1 cells were resuspended in 1 ml media with 25 µl of the concentrated virus. 24 h after transfection, fresh media with 2 µg ml$^{-1}$ puromycin was added and selection carried out for 48 h. Selected cells were passaged four times and single cells with GFP signal were FACs sorted. The PF1 clone was selected from this line after subsequent FACs analysis where both GFP and Kusabira orange 2 signal was analyzed.

### Hoechst and FACS analysis

Two drops of Hoechst 33342 Ready Flow Reagent from Invitrogen was added to 2 × 10$^6$ cells and incubated for 15 min. Cells were spun, resuspended in Miltenyi Biotec FACs buffer and assayed using the BD FACSAria Fusion 3. Green fluorophores were excited at 488 nM with emission at 530 nM, whereas orange fluorophores were excited at 561 nM with emission at 586 nM. Fluorophores in Phenomex Lightning were detected using the FITC and TRED excitation/emission filters in the Cell Analysis Suite. Hoechst staining was measured at 375 nM with emission at 450 nM. To determine FACs signal overlap between fluorophores and Hoechst, 10,000 cell measurements were read into R, Hoechst signal was split into 100 bins and cell cycle fluorophores scaled from 1 to 100 and intensities were compared.

### UV treatment conditions

To determine UV intensity, 2.5 × 10$^5$ PF1 cells in 500 µl media were plated in six-well dishes. UVC treatments used the Analytik Jena crosslinker (model CL-1000, 254 nm) at exposures of 5,000–30,000 µJ/cm$^2$ (Extended Data Fig. 2a). After treatment, 2 ml fresh media was added and cells cultured for 3 days. Cell numbers were counted in triplicate using the Countess 3 from Thermo Fisher. UV at 5,000 µJ/cm$^2$ was used given roughly half of the cells proliferated post treatment.

### P388D1 genomic DNA extraction

Cells were pelleted (5 min @ 500 × $g$), washed with 1 ml PBS and resuspended in 200 µl PBS. DNA was extracted with the Qiagen DNeasy kit. After resuspension in 200 µl PBS, 20 µl proteinase K and 200 µl buffer AL were added, briefly vortexed and incubated for 30 min at 56 °C with rotation @ 400 rpm. Next, 200 µl 100% EtOH was added, and the lysate was spun through a DNeasy mini spin column at 8,000 × $g$ for 1 min. The column was washed with 500 µl AW1 and 500 µl AW2, then spun at 18,000 × $g$ for 3 min. To elute DNA, 100 µl buffer AE was added, the column was incubated for 5 min at 37° and spun for 1 min at 8,000 × $g$.

### Whole-genome library construction and sequencing

Genomic DNA size and quality was assayed using the NanoDrop and Agilent 4200 TapeStation. Libraries were prepared using 100–500 µg DNA and the NEBNext Ultra II kit with Unique Dual Index primers for Illumina. Enzymatic fragmentation was carried out for 15 min instead of 5. Libraries were amplified between four and six cycles using the NEBNext UDI primers (article E6440). Library size and molarity was determined using the TapeStation system and libraries pooled at a concentration of 2 µM. Paired-end 100 bp sequencing was performed using the NextSeq 2000 and NovaSeq platforms.

### Total RNA extraction and library preparation

Total RNA was extracted from three replicates of 10$^6$ PF1 cells using the RNeasy Plus Mini kit from Qiagen. Total RNA quality and quantity was assayed using the RNA ScreenTape on the Agilent TapeStation system. 100 ng total RNA from each replicate was processed with the TruSeq Stranded total RNA with Illumina Ribo-zero Plus RNA depletion protocol. Libraries were quantified using the TapeStation High Sensitivity D5000 ScreenTape and QuBit dsDNA High Sensitivity kit. Libraries were Sequenced on the NextSeq 2000 with 50 bp paired-end reads (100 cycle P2 chemistry).

## Omni-ATAC library preparation

Omni-ATAC[63] libraries were carried out with slight modifications[64,65]. Pellets of 50,000 cells were washed in 100 µl PBS and resuspended in 50 µl lysis buffer (10 mM Tris-HCl pH 7.5, 10 mM NaCl, 3 mM MgCl$_2$, 0.1% NP-40, 0.1% Tween-20 and 0.01% Digitonin). After 3-min incubation on ice, 1 ml wash buffer was added (10 mM Tris-HCl pH 7.5, 10 mM NaCl, 3 mM MgCl$_2$, 0.1% Tween-20) and centrifuged for 10 min at 500 × g at 4 °C. Nuclear pellets were resuspended in 50 µl transposition mix (25 µl 2X Tagment DNA buffer (Illumina), 16.5 µl PBS, 0.5 µl 10% Tween-20, 0.5 µl 1% Digitonin, 2.5 µl Tn5 Tagment DNA enzyme (Illumina) and 5 µl H$_2$0). Nuclei were incubated 30 min at 37 °C, DNA cleaned using the Qiagen MinElute Reaction Cleanup kit and eluted in 10 µl elution buffer. Libraries were amplified for five cycles using the universal primers and barcoded primers A2.1, A2.2 and A2.3 (ref. 65) for replicates 1, 2 and 3, respectively. An additional 8 cycles were determined using qPCR[65] and carried out. Libraries were sequenced on the NextSeq 2000 platform using paired-end 50 bp reads (100 cycle kit, P2).

## Whole-genome and ATAC alignment and filtering

Raw reads were trimmed with the TrimGalore[66] using Python 3.6.1 (ref. 67) and the –stringency 3 flag. WGS and ATAC reads were mapped with bowtie2 (ref. 68) using the –end-to-end and –maxins 1,000 flags. PF1 samples used the mm10 reference genome while a dual-hybrid N-masked reference genome (see below) was used for F1 tumors. Bam files were processed with samtools v1.102 (ref. 69) matefix and markdup tools and filtered using the 0 × 2 flag. The analyzable fraction of the genome was determined by counting reads in 1-kb windows for triplicate negative control samples using bedtools 2.24.0 (ref. 70). Regions with ≥1 standard deviation from the mean in two of the three replicates were excluded corresponding to 10.2% of the genome. X chromosome counts were doubled in this analysis to account for the single copy.

## Dual-hybrid N-masked reference for WGS F1 tumor haplotype discrimination

The SNPsplit[47] program was used to create a dual-hybrid reference where germline SNPs from both C3H and *Mus castaneus* were replaced with 'Ns' in the mm10 reference using the command:

*SNPsplit_genome_preparation –vcf_file mgp.v5.merged.snps_all. dbSNP142_UCSC.vcf –strain C3H_HeJ –reference_genome BSGenome_ mm10.fasta –full_sequence –nmasking –dual_hybrid –strain2 CAST_EiJ*

Germline SNPs in reference to the mm10 build were downloaded from the Mouse Genomes Project[46,71,72]. Trimmed WGS reads were mapped, matefixed and duplicates marked and filtered as above. Reads were then split using the SNPsplit command with the –conflicted and –paired flags.

## Mutation calling and filtering

A Strelka2 (ref. 73) pipeline was used to call mutations from WGS reads in both PF1 samples and F1 tumors. F1 tumors used the SNPsplit reference above, whereas PF1 cells used the standard mm10 reference. Manta was first run on both tumor/normal and cell line/UV treated cells to flag structural variants before Strelka2 mutation calling. Both Strelka2 and Manta used default parameters. Mutations were originally processed with bcftools[69] for the PASS flag, and then the GATK[74] CalculateSNVMetrics walker was used to further filter mutations. Mutations were removed if:

*Variant*AlleleCount < 4, VariantAlleleCountControl > 1, VariantMapQualMedian < 40, MapQualDiffMedian < −5.0, MapQualDiffMedian > 5.0, LowMapQual > 0.05, VariantBaseQualMedian < 30, (VariantAlleleCount >= 7 & VariantStrandBias < 0.05 & ReferenceStrandBias >= 0.2), DistanceToAlignmentEndMedian < 10, DistanceToAlignmentEndMAD < 3

For F1 tumors, mutations overlapping germline SNPs were also removed from the analysis.

## PF1 ATAC-seq data processing

Aligned ATAC reads were used as input for MACS2 (ref. 75) to call peaks with the flags -f BAMPE -g mm –nomodel –nolambda –keep-dup-all –call-summits -B -q 0.01. Peaks from individual replicates were converted to GRanges objects in the R environment and merged using the reduce function of the GenomicRanges[76] package v1.52.1. Read counts within merged peaks were calculated using the qCount function of the QuasR[77] package v1.40.1.

## PF1 RNA-seq data processing and analysis

Transcript abundances were quantified using Kallisto[78] v0.46.0 with the –bias and –rf-stranded flags and the Gencode M25 transcript release. To assign tags per million to a single gene instance, transcripts were split based on shared Entrez gene ID. Gene IDs with transcripts on more than one chromosome, transcripts on both strands, transcripts with no Entrez gene ID or overlapping genes with the same alias were removed. To determine a gene model, ATAC signal was calculated within a 1.5-kb window around each annotated TSS (−1,000, +500). Gene starts were selected from the transcript with maximum ATAC signal, or in the case of no-expression, the longest isoform was used. All transcripts from an Entrez gene ID with unique signal from quantification with Kallisto were summed, and the end of the gene model determined by the longest transcript with quantified reads. To bin genes based on transcription level, genes were first filtered to be at least 1 kb long and 1 kb away from the nearest neighbor to prevent confounding signals from repetitive small genes and genes in direct proximity, resulting in 19,091 genes total. A pseudocount of 0.1 was added to the mean TPM of each gene and log$_2$ converted. Genes with a value of 0 or below were designated as bin 1 and represented 10,657 genes. The remaining 8,434 'expressed' genes were binned into three quantiles, resulting in 2,812, 2,811 and 2,811 genes in bins 2, 3 and 4, respectively. To visualize RNA reads (Fig. 4a), RNA was aligned to the mm10 reference using STAR[79] 2.7.10b and the ENCODE parameters specified in the STAR manual.

## HEK293 Flag-OGG1 ChIP and total RNA-seq processing and analysis

Published Flag-OGG1 ChIP[41] and Ribo-Zero total RNA[80] data were downloaded from GEO accession numbers GSE89017 and GSE76496, respectively, and trimmed with TrimGalore. RNA was mapped to the T2T-CHM13 version 2.0 (ref. 81) reference using STAR[79] as above, whereas Flag-OGG1 data was mapped using bowtie2 (ref. 68) version 2.3.5.1 and unique reads were retained. A GFF3 RefSeq file was downloaded from the UCSC genome browser, and genes were filtered to have gene_biotype = 'protein coding' and extra_copy_number = '0', resulting in 19,776 genes. To avoid neighboring interference and noise from very small gene bodies, genes were further filtered to be at least 5 kb in genomic length and at least 5 kb away from the nearest neighboring gene (5 kb profile flanks +1 kb buffer), resulting in 13,766 filtered genes. Tags per million (TPM) were tallied for each gene, and genes with greater than 0 TPM were quantile binned, resulting in four bins of 4,094 for nontranscribed or very low transcribed genes and 3,224 genes each for low, medium and high categories. Each gene body was divided into 100 equal tiles using the tileGenome function of the GenomicRanges package, and each 5-kb flank was divided into 10 tiles of 500 bp each. Reads per kilobase were tallied for each tile. Mean log$_2$(counts) for each tile in a respective genomic bin was calculated to produce metaplots of signal (Extended Data Fig. 4h).

## Multiallelic analysis

Tandem mutations were identified using the GenomicRanges distanceToNearest function, selecting those with an intermutation distance of 0. To calculate VAFs for each base in a tandem mutation, reads were extracted from the relevant PF1 genome using Rsamtools[82] v2.12.0, whereby each read in the calculation had sequence information for both bases. Mutliallelic sites were selected on the criteria that two alternative

alleles with at least 3 unique reads with information at both bases were present. To determine mutation order, alleles at each multiallelic site were ordered based on VAF, with the higher VAF allele assumed to be the first mutation at that site. To confirm VAF bias for the second C in tandem CC > TT mutations, we used a grepping approach. For each multiallelic site, we used 10 bp of sequence on each side of the tandem site (22-base-long sequence as query) for all possible alleles and their reverse complement (8 in total per multiallelic site). To compare with the alignment data, we filtered sites where one of the alternate alleles had an exact match to another 22 bp sequence in the genome. We further filtered for regions with 0 matches that occurred because of single base pair changes in the genome that precluded an exact string matching event. In total this resulted in 237 tandem CC > TT mutations to compare VAFs for each base in the tandem mutation (Extended Data Fig. 3).

## RL20 metric
The rl20 metric was carried out as previously described[14]. In short, run lengths of relevant mutations (eg, C > T or G > A for UV) were calculated on a chromosomal basis using the rle (run length encoding) function in R and runs were ordered by decreasing size. The smallest run length in the top 20% of this list was set as the run length for that particular genome and informative mutation type. The significance of seeing such a run length given equal probability of either mutation orientation was calculated using a two-sided Wald-Wolfowitz runs test with the runs. tests function in the R package randtests[83] v1.0.1.

## Mutational phasing
To compare mutational phasing between samples, the genome was tiled into sliding 10-Mb bins with a 100-kb step using the GenomicRanges[76] slidingWindows function. Bins with 95% mappability were retained, as determined using the function mappabilityCalc in the Repitools[84] v1.46.0 R package. Overlapping mutations of either stranded-orientation, for example C > T and G > A with UV, were assigned a 1 or −1 respectively. The average of this number for each bin represented the phasing of that bin.

To simulate a mutation phasing distribution assuming a lesion segregation phenotype with these bins, we created three sets of Bernoulli trials in a 1:2:1 proportion. Using UV as an example, these sets represented C > T phased segments, mixed segments and G > A phased segments, respectively (Extended Data Fig. 5). Each mutation within the bin was thus a trial, while bins represented sets of trials. The 1:2:1 proportion was used as the expected ratio under the Hardy-Weinberg[85] assumption, given each mitotic sister inherits the mutation result of two strands non-selectively in mitosis. We next set C > T mutations as a success while G > A mutations were a failure. The probability of success in mixed bins of the genome was thus set to 50%, resulting in an equal representation of both mutation types.

For phased bins, under perfect circumstances in the framework of this model it would be assumed the probability of success is 100% in a C > T phased segment, and 0 in a G > A phased segment. This accuracy was not reflected in the genome, as even in phased segments of F1 resolved genomes and the singular X chromosome, roughly 5% mutations are out of phase (Extended Data Fig. 6).

To calculate the out-of-phase rate in our PF1 cell line data, we focused on the mutations shared between mitotic sister cells at the time of penning. Our reasoning was that while C > A mutations made up the majority of the population, on average 31% of this number were C > T or G > A and thus indistinguishable from true UV mutations. This meant that given the number of sister-specific ROS mutations, between 5% and 16% of this number are false positive UV mutations (Extended Data Fig. 5). Probability of success was adjusted to reflect this fact. More specifically, if 12% of C > T mutations in a particular genome are assumed to be background, the probability of success was shifted by 6% as these 'false positive' UV mutations would be assumed to be incorrectly phased half the time.

For a 10-Mb bin to be considered in the model, at least 10 mutations needed to be present in that bin, which equated to 13,971 bins for UV and 16,337 bins for ROS. To create an exhaustive population of these segments and establish an ideal distribution, we carried out 100 fold more Bernoulli trials than were present in all 14 genomes, which equated to 139k for UV and 163 K for ROS. This number was chosen as it is two orders of magnitude larger than the actual population. We also carried out a set of trials with the same amount of data points actually represented in the data (13.9k and 16.3k), to directly compare the distributions qualitatively and subsequently with a qq-plot of the resulting distributions.

## SCE
To delineate mutations specific to the recombined strand, we first identified the phased and unphased segments on each side of an SCE site. This was done by taking the absolute value of the skew, which scaled from −1 to 1. The mixed segment was determined as the smaller absolute value of skew. After identifying the skewed segment, we identified the polarity of the adjacent segment by asking if the skew was greater or less than 0, in the case of UV meaning it was either a C > T or G > A phased segment respectively. Finally, mutations in the mixed segment with the opposite orientation of the adjacent skewed segment were identified as recombined strand mutations. In contrast, mutations in the mixed segment with the same polarity as the adjacent skewed segment were noted as non-recombined strand mutations.

To profile mutation density for both strands and the mixed regions around SCE sites, a 1-Mb sliding window approach with 100-kb step size was used as above, covering in total 20 Mb centered on the SCE site. Mutation rate was reported as the number of mutations/megabase (mu/Mb).

## Mutation rate calculation
Mutation rates were calculated as previously reported[14]. To account for genomic representation in mutation rate, each mutation type (ie, C > T) in addition to bases immediately adjacent to the mutation site were first summed, creating a trinucleotide vector of 192 unique mutation instances. This was folded into a vector of 64 unique mutations by combining identical trinucleotide contexts, where only a different alternate base was observed. The number of trinucleotide mutation instances was then divided by the total number of possible trinucleotides in that window. The weighted mean of this number for all trinucleotides was then calculated, with the weights being the relative representation of that specific trinucleotide in either the window of interest or the whole genome, depending on the comparison. This number was then multiplied by $10^6$ to represent mutations/megabase. For mutation specific rates such as UV and ROS, the same was applied but trinucleotides and identities were subsetted to only reflect these mutation types.

## Mutation signatures
To identify mutation signatures, trinucleotide sequences centered on each mutation were first reverse complemented if the reference base was either A or G. This created a vector of length 96, representing all mutations in the context of either a C or T reference base. The number of that mutation type was divided by the total mutations for that sample to depict a frequency of each mutation identity in the population. To compare our observed mutation signatures to previously identified ones, we downloaded SBS signatures from the COSMIC database version 3.2 for the mouse genome reference GRCm38. This consisted of 79 total signatures, 19 of which were filtered out given evidence of possible sequencing artefacts. Using our frequency scaled signatures defined above, we compared the cosine similarity of the 96 length vectors for each sequenced genome to the 60 filtered COSMIC signatures using the cosine function in the lsa[86] package.

## Transcription-coupled repair analysis
To compare mutation rates to transcription output we used the gene models and bins defined above. In the case of UV mutations, template

strand damage was determined as a C > T mutation in minus strand genes and a G > A mutation in plus strand genes. Conversely, C > T mutations in plus-strand genes and G > A mutations in minus strand genes were defined as non-template mutations. The same logic was applied to ROS mutations, meaning G > T mutations on minus-strand genes and C > A mutations on plus strand genes were determined to be template mutations. The inverse again were designated as non-template mutations. Mutation rates were calculated as above, with mean weights calculated using the trinucleotide representation of the whole genome. Stranded mutation rates were multiplied by 2 and then divided by the genome average, to represent rate relative to the genome.

### R programming environment
Analyses in the R programming language were performed in Rstudio[87] using R[88] v4.3.0. Additional packages used in analysis and visualization not explicitly cited in the text include RcolorBrewer[89] v1.1–3, pheatmap[90] v1.0.12, NMF[91] v.0.26, TxDb.Mmusculus.UCSC.mm10.knownGene[92] v3.10.0, BSgenome.Mmusculus.UCSC.mm10 v1.4.3, rtracklayer[93] v1.60.0, ggplot2 (ref. [94]) v3.4.4, vcfR[95] v1.14.0, scales[96] v1.2.1, regioneR[97] v1.32.0, Gviz[98] v1.44.2, bsub v1.1.2, viridis[99] v0.6.4, QuasR[77] v1.40.1, maptools[100] v1.1–8, apcluster[101] v1.4.11, mixtools[102] v2.0.0 and lsa[86] v0.73.3.

### Statistics and reproducibility
No statistical method was used to predetermine sample size and no data were excluded from the analysis. The experiments were not randomized and the investigators were not blinded to allocation during experiments and outcome assessment. Pearson's median skewness coefficient was calculated from each VAF distribution in R as (3×(mean-median))/standard deviation. Additional statistical tests and calculations are as described in the relevant Methods sections or figure legends.

### Reporting summary
Further information on research design is available in the Nature Portfolio Reporting Summary linked to this article.

### Data availability
Fastq files for the WGS, RNA-seq and ATAC-seq data produced for this paper can be downloaded from Sequence Read Archive under the accession number PRJNA934746. Processed files, including mutation calls, TPM counts and ATAC peaks used in the analysis, have been deposited in GEO under the accession GSE230579. HEK293 Flag-OGG1 ChIP-seq data were downloaded from the GEO accession GSE89017, whereas HEK293 Ribo-Zero total RNA data were obtained from GEO accession GSE76496.

### Code availability
R scripts to reproduce all main figure panels can be downloaded from GitHub (https://github.com/odomlab2/Single-Mitosis-LSE)[103] with https://doi.org/10.5281/zenodo.10786189 (ref. [104]).

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

## Acknowledgements

This work was supported by the Deutsches Krebsforschungszentrum (DKFZ), the European Research Council (grant 788937 to D.T.O.), with additional support from the MRC Human Genetics Unit core funding program grant MC_UU_00035/2 to M.S.T. S.J.A. is supported by core funding from the MRC Toxicology Unit (RG94521). We would like to explicitly thank the following colleagues: F. Connor from the University of Cambridge, UK for assistance in the F1 mouse DEN experiments; E. Boga, T. Wagner, P. Schmidt, R. Embacher and S. Eichmüller for assistance with the Berkley Lights Lightning platform; N. Glaser and the DKFZ sequencing core for assistance in DNA sequencing; S. Schmitt of the DKFZ FACs core for assistance in FACs analysis; and A. Feldmann from the DKFZ for critical input on the manuscript.

## Author contributions

P.A.G., M.S.T. and D.T.O. conceived the study and designed experiments. P.A.G., H.B. and A.S. performed UV mutagenesis, cell-splitting experiments, WGS library preparation, ATAC-seq and RNA-seq experiments. S.J.A. and C.E. performed mouse experiments. M.B. performed supporting experiments. P.A.G. and M.S.T. designed and implemented computational analysis with support from D.T.O. D.T.O. and M.S.T. supervised the work. P.A.G., S.J.A., D.T.O. and M.S.T. wrote the manuscript, with input from all other authors.

## Funding

## Competing interests

The authors declare no competing interests.

## Additional information

**Extended data** is available for this paper at https://doi.org/10.1038/s41588-024-01712-y.

**Correspondence and requests for materials** should be addressed to Martin S. Taylor or Duncan T. Odom.

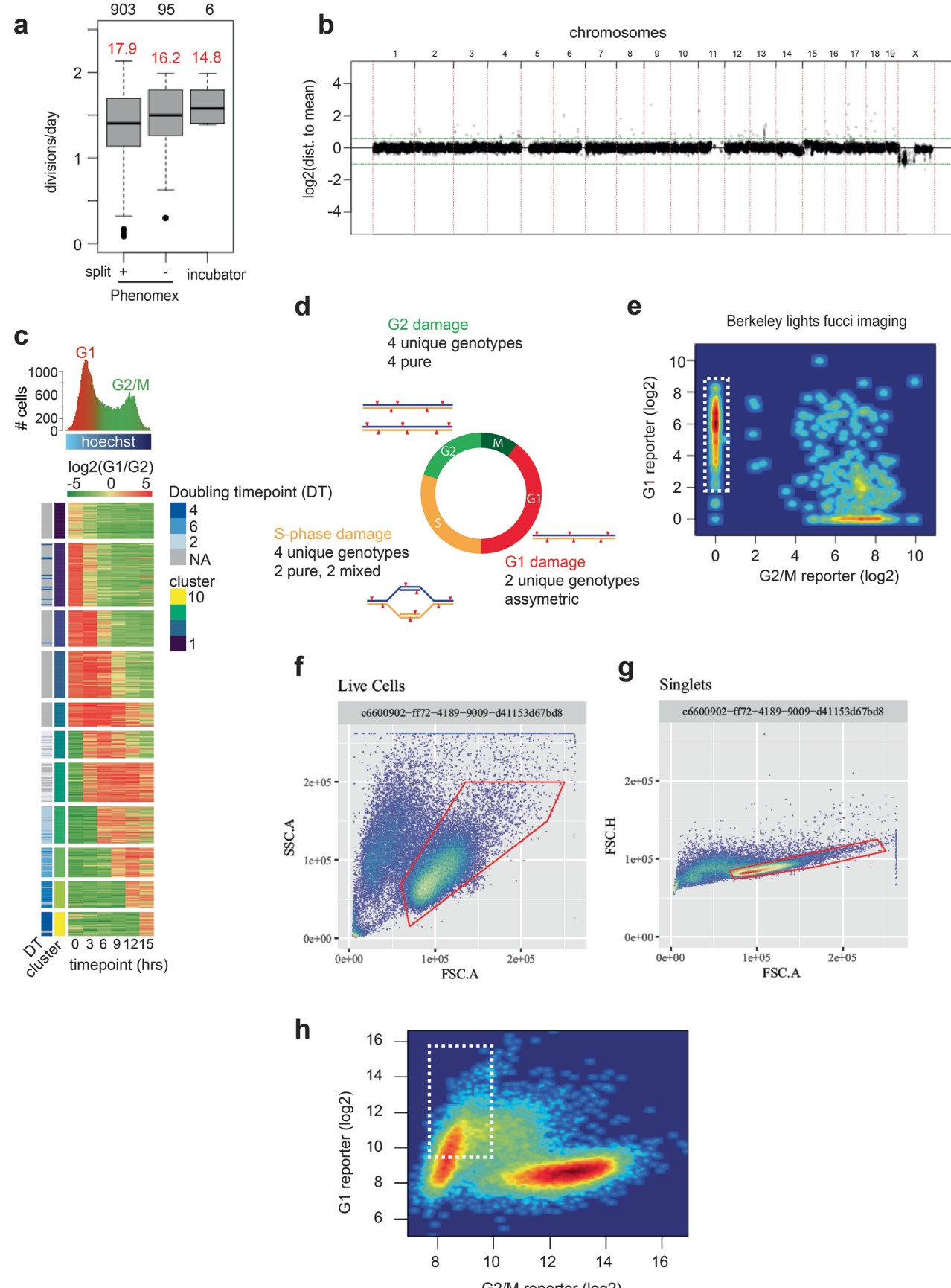

**Extended Data Fig. 1 | See next page for caption.**

**Extended Data Fig. 1 | Genomic stability and cell cycle determination in the PF1 line. a**, Divisions per day on the Phenomex instrument and in cell culture. Rates for cells split after a single mitosis are noted. Numbers above boxes represent the total number of rates measured, for Phenomex this reflects pens, for incubator cells it is individual wells. Red numbers represent the mean doubling time in hours. Boxplot elements are as described in Fig. 3c, albeit without notches. Proliferation measurements from the Phenomex platform were taken on cells that proliferated post UV treatment. **b**, Copy Number analysis, showing diploid content for most of the genome. Reads were counted in 10 kb bins, and the y axis represents log2(distance to mean across all bins). Red vertical lines demarcate chromosome boundaries, and green horizontal lines represent counts expected for a single copy number gain or loss. **c**, Top: DNA content (x axis = Hoechst intensity) as a function of green and red fluorescence. Histogram bins have been colored by scaled log2(red/green) for each cell. Bottom: FUCCI fluorophores imaging over time. 357 cells on one chip were affinity propagation

clustered based on FUCCI across all 6 timepoints. Color scale is noted above and is identical to the Hoechst histogram color scheme. Doubling time point (DT) is indicated by the second annotation column scaling from early replicating timepoints (gray) to later replication timepoints (dark blue). Timepoints are 3 hour intervals and noted below each column. **d**, Fluorophore signal per cell cycle and theoretical effect of ploidy on mutation patterns for pulse mutagenesis (UV). Cells in S-phase would have intermittent lesion segregation patterns, while cells with duplicated DNA (G2/M) would not show lesion segregation patterns after a single mitosis. **e**, Scatter of scaled G1 (red) and G2/M (green) signal directly after penning for 1120 cells measured on the Phenomex platform. Cells to split are indicated by the white dashed box. **f**, Gating Live cells with FSC-area by SSC-area. **g**, Singlet determination by FSC-area by FSC-height. **h**, Fluorophore intensity for G1 fluorophore (FITC) and G2/M fluorophore (yellow-green laser) as measured by FACs. White box denotes FITC positive cells that were single-cell sorted to establish the PF1 line.

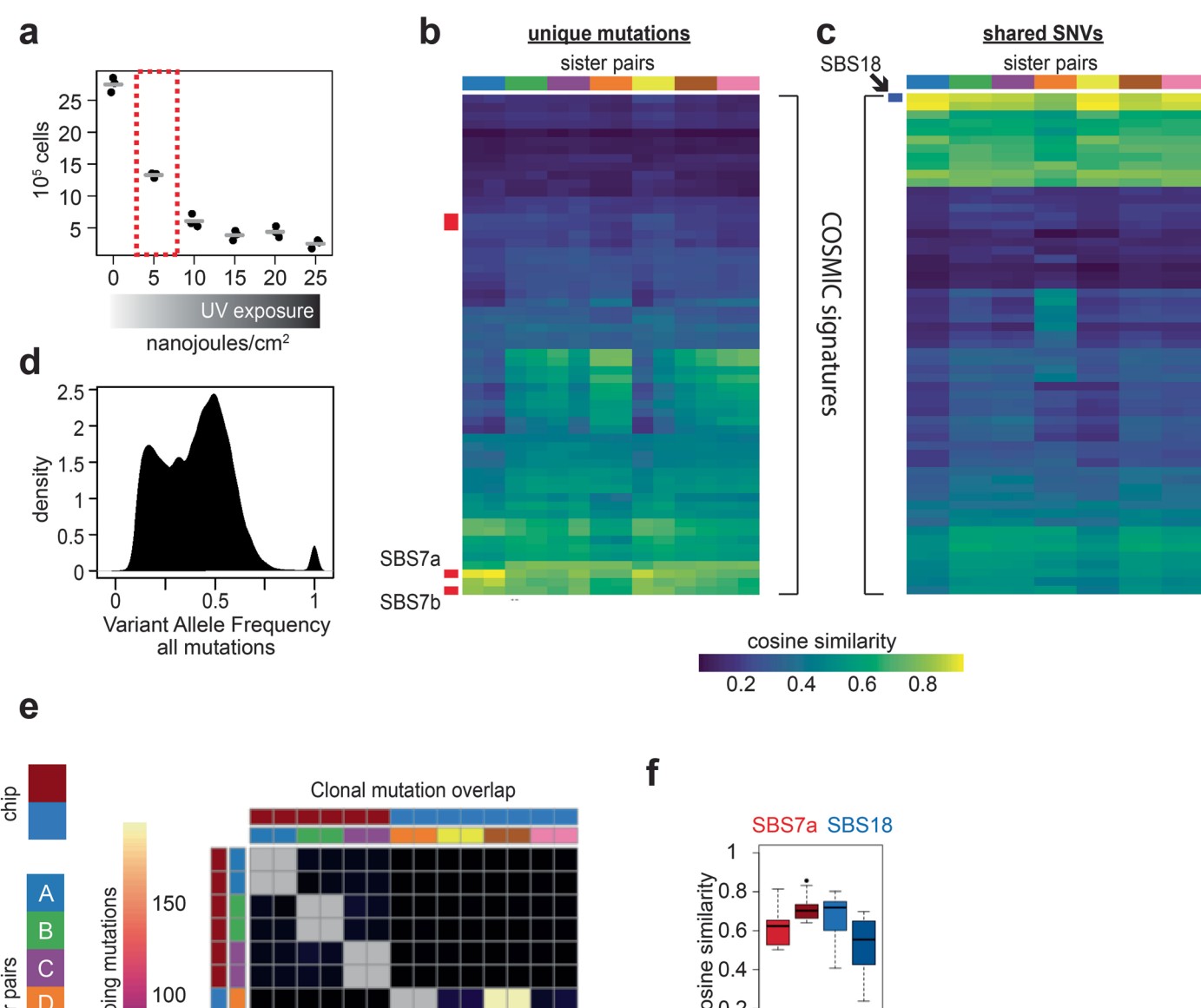

**Extended Data Fig. 2 | UV treatment, mutation signature determination and QC of WGS data. a**, Determination of UV treatment intensity and effect on viability 3 days post exposure. Intensity used to induce UV mutations is denoted by the red dashed box. Y axis represents viable cells, X axis represents UV intensity in nanojoules/cm². **b**, Cosine similarity between sister-unique mutations and 60 annotated SBS signatures in the COSMIC database. The four signatures attributed to UV damage are noted by the row annotation in red. Mitotic sister pairs are noted by the column annotation at the top of the heatmap. **c**, Same as in C but for mutations shared between mitotic sisters. ROS signature is noted by the annotation row in blue. **d**, Distribution of VAF for all mutations across all sisters. **e**, Heatmap displaying the number of overlapping mutations

between clones (see Fig. 2c, bottom), sister-shared mutations are grayed out to demonstrate clonal sharing. Chip annotation bar depicts two independent Phenomex chips (independent splitting experiments). Sister pair color annotations are as shown in **b** and **c**. Sisters with higher clonal mutation overlap stem from two individual clones from a single splitting experiment. **f**, Similarity of mutation signatures to UV (SBS7a, red) and ROS (SBS18). Each pair of box plots represents the similarity of all 14 individual sisters to the respective signature when all mutations are considered (light red and light blue) or only mutations unique to that sister (dark red and dark blue). Boxplot elements are as described in Fig. 3c, albeit without notches.

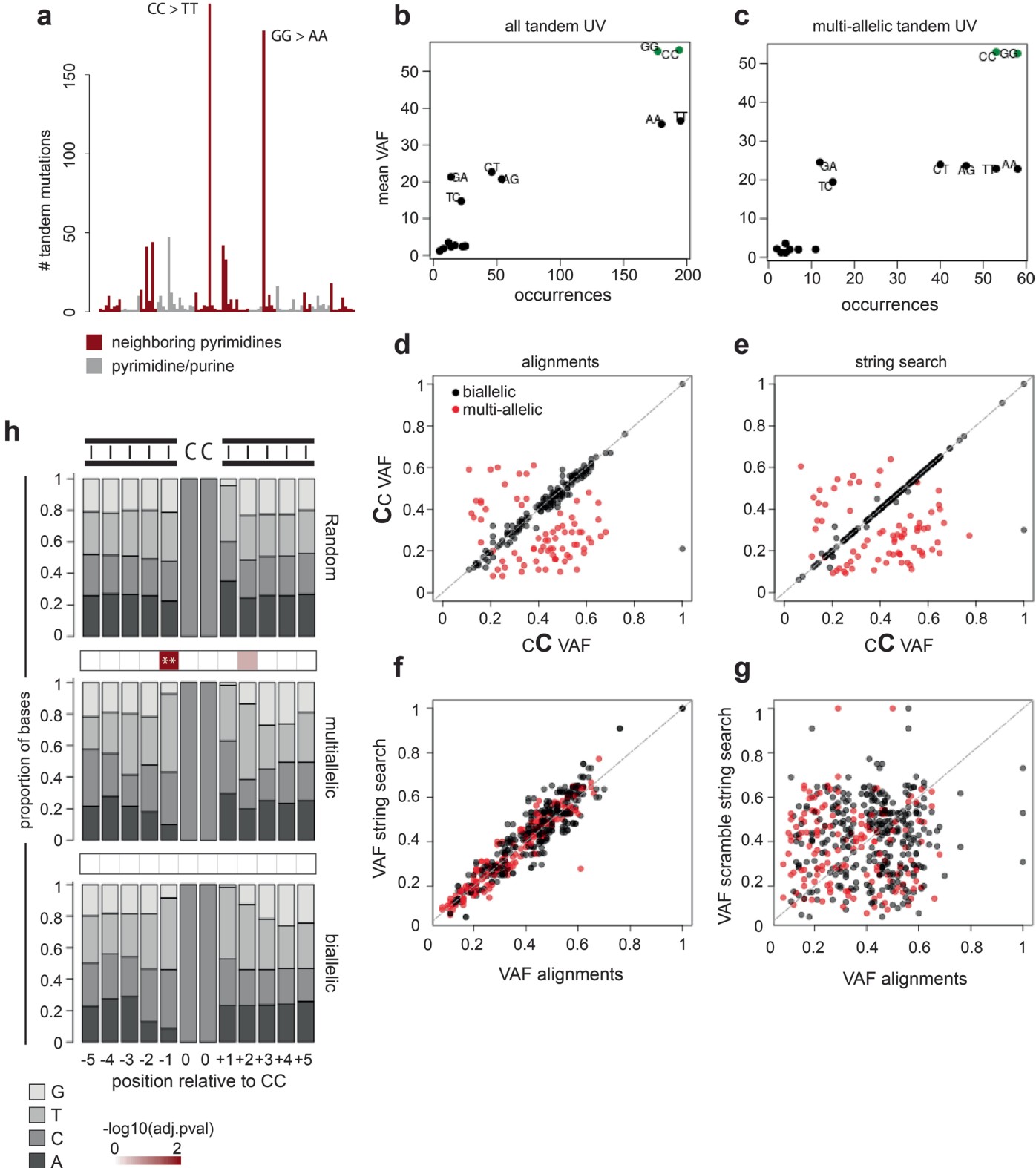

**Extended Data Fig. 3 | See next page for caption.**

**Extended Data Fig. 3 | Unrepaired UV lesions can create multiallelic variation. a**, Barplot of counts for all tandem mutation identities (929 total dual mutations, 94 total categories). Red bars represent reference alleles where two pyrimidines are adjacent to each other (ie, CC, CT, etc.) while gray bars represent purine/pyrimidine hybrids (ex CG). Canonical CC > TT UV mutations, and its reverse complement are noted (373 in total). **b**, Scatterplot depicting number of times an allele was seen at a tandem mutation, and the average VAF for that allele when it is detected. **c**, Same as in (b) but only for multiallelic tandem mutations. **d**, Comparison of VAFs for each cytosine in a tandem mutation, calculated from reads where information for both bases is contained. Note how the 3' CC is more likely to be a mutation seen in multiallelic UV sites. 237 of the 373 sites are represented after filtering for dual mutations that can be interrogated using the string searching approach (see Methods). **e**, The same as in (d), but instead of using alignment information directly, sequences representing each UV mutation allele were detected by string searching raw sequences in the corresponding genome (see Methods). **f**, Scatterplot of VAFs calculated from alignments (d) with VAFs calculated from string searching sequences from raw data (e). **g**, Same as in (f), except the identity of sequences for VAFs calculated from string searching (Y axis in f) have been scrambled to depict no relationship. **h**, Sequence context surrounding CC dual mutations. Surrounding base identities have been converted to proportions. The identity of each base is shown in the key at bottom. The heatmap between any two bar plots displays the -log10(Benjamini-Hochberg adjusted p-value) from a Fisher's exact test between base counts at the position (two-sided alternative hypothesis). Positions with a p-value below 0.005 are noted by '**'. (*top*) The average of 100 random sampling events of 111 CC dinucleotides in the mouse genome. (*middle*) Identities of surrounding bases for all multiallelic sites (n = 111). (*bottom*) Identities of surrounding bases for all biallelic sites (n = 272).

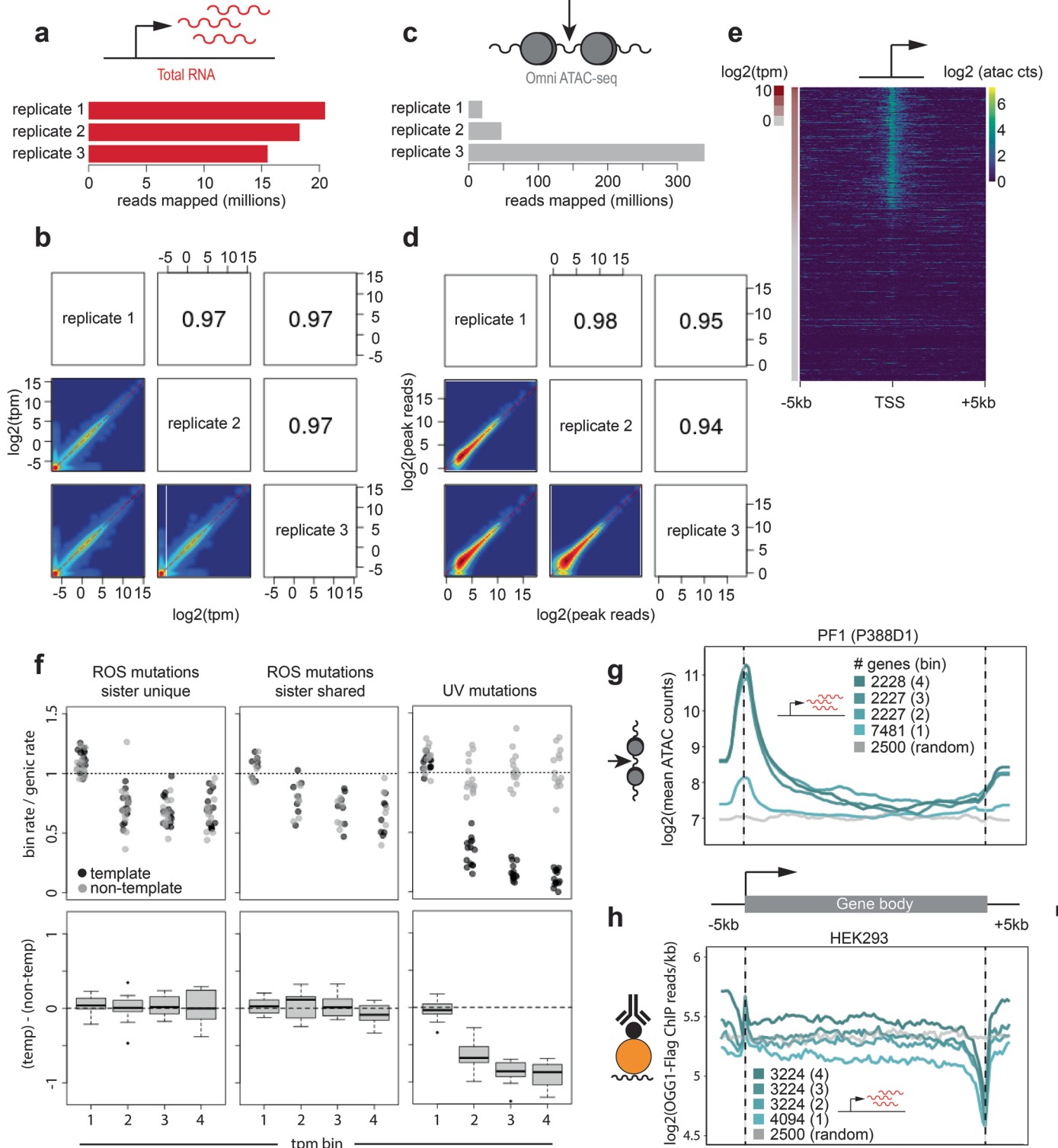

**Extended Data Fig. 4 | See next page for caption.**

**Extended Data Fig. 4 | Transcriptome and accessibility profiling in the PF1 cell line. a**, Number of unique reads mapped (in millions) per library for triplicate total RNA-seq replicates. **b**, Pairwise scatters of RNA measurements for all annotated mouse genes. Values are shown as log2(TPM + 0.01). Upper panels represent the Pearson correlation coefficient for the respective scatter. **c**, Same as in A but for unique reads in triplicate ATAC-seq samples. **d**, Pairwise scatter of reads in merged peaks across 3 ATAC-seq replicates. Axes represent log2(reads per kb + 1). **e**, Heatmap of ATAC-seq counts in a 10 kb window surrounding transcription start sites. Rows are ordered by TPM from RNA-seq data in (a), and represented as the annotation column to the left of the heatmap. **f**, Upper: stripchart of template (black points) and non-template (gray points) mutation rates divided by the total genic mutation rate for all 14 genomes. Point clusters represent genic bins as described in Fig. 4. From left to right, ROS mutations

unique to each sister cell (14 points per bin), ROS mutations shared between sisters (7 points per bin), and UV mutations (14 points per bin). Lower: Boxplot of template - non-template rate for all 14 genomes, considering the mutations as for the stripchart panels above. Boxplot elements are as in Fig. 3c without notches. **g**, Average ATAC signal over gene bodies. Genes at least 5 kb in length were first binned based on TPM from low (1, light blue) to high (4, dark blue), and additionally 2500 coordinate shuffled gene positions (gray) were taken as a negative control. Gene bodies were divided into 100 tiles. Additionally, a window of 5 kb was added flanking the TSS and TTS. Reads were counted in all genic tiles, summed by genic bin, and scaled to reads per kb of genomic representation. **h**, Genic signal for Flag-OGG1 ChIP data in HEK293 cells[42]. Transcriptional binning and gene body tiling were performed as in panel (g), and numbers of genes per bin are shown as in (g).

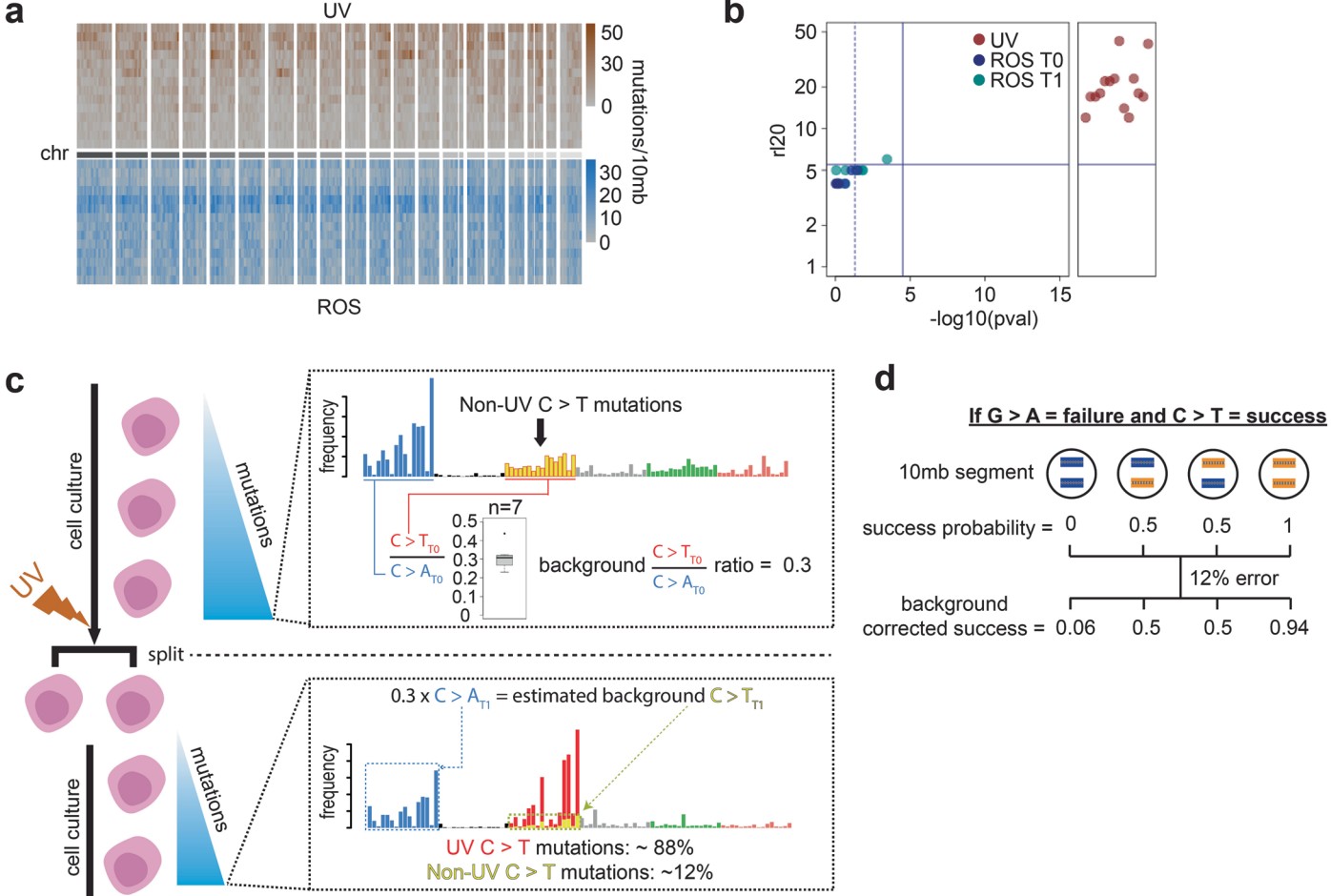

**Extended Data Fig. 5 | Mutational phasing for UV damage in mitotic sisters.**
**a**, Mutation density for UV (upper) and ROS (lower) across all chromosomes. Heatmaps represent 10 mb sliding genomic windows with a 1 megabase step. Mutation density in windows is represented as mutations per 10 mb window. Each row is a single sister genome and rows are sorted by total UV mutation counts from highest to lowest. **b**, rl20 analysis (see Methods) as in Fig. 4a, but distinguishing between ROS mutations shared between mitotic sisters at time point 0 (dark blue) and ROS mutations unique to each individual sister, acquired after the first division (light blue). Red dots represent UV mutations. The y axis depicts the longest set of runs for a single mutation type, accounting for 20% of all informative mutations. The x axis is the significance in -log10(p-value) for seeing such a run length given random assignment to strands, calculated using a two-sided Wald-Wolfowitz runs test. **c**, Schematic depicting determination of background C > T mutation when modeling phasing for UV damage using

Bernoulli trials. Upper box: Cells accumulate Non-UV C > T mutations in culture (yellow bars with red border) before UV damage. The ratio of C > T/ C > A for all 7 sister pairs is shown in the boxplot inset (Boxplot elements are as described in Fig. 3c, albeit without notches), and the average is ~ 0.3. Lower box: Total C > A mutation counts unique to each sister is multiplied by 0.3 to estimate the amount of total background C > T mutations (overlaid yellow bars). This background C > T estimate is then divided by the total C > T mutation counts to estimate the error adjustment for phased Bernoulli trials, which has a mean of 11.8% and ranges from 5% to 16% depending on UV total mutations. **d**, Error rate is used to adjust success or failure probability for completely phased segments. An example error rate of 12% is shown, whereby each phased segment probability is adjusted by 6%, as half of these background mutations will be randomly out of phase.

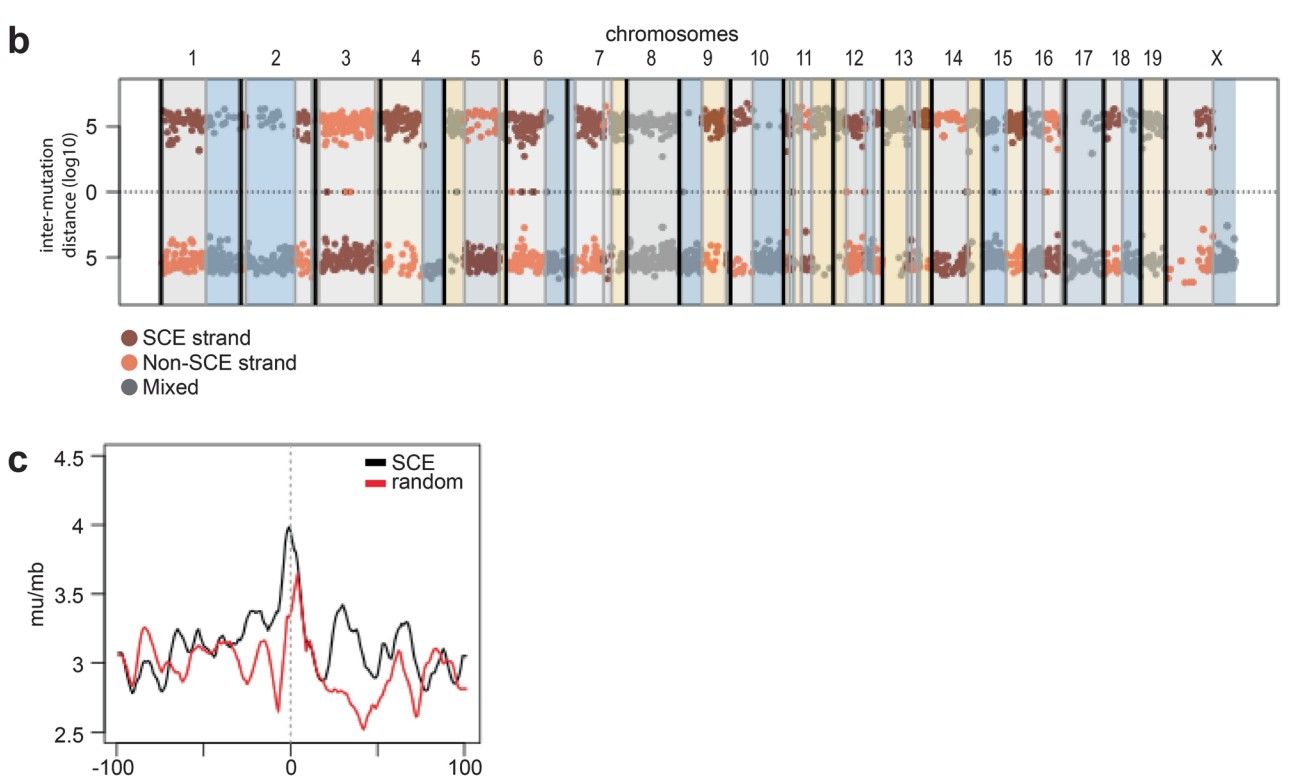

**Extended Data Fig. 6 | See next page for caption.**

**Extended Data Fig. 6 | Acute, single pulse damage reveals asymmetric lesion segregation patterns between mitotic sisters. a**, Segmented heatmap for UV (upper) and ROS (lower) induced mutations for 7 pairs of mitotic sisters. Model of mutation phasing for a single burst event (UV, upper right) that results in a lesion segregation phenotype, as opposed to chronic, low mutation rate (ROS, lower right). **b**, Plot of UV mutations with reference C bases in the upper and reference G bases in the lower halves respectively. Lightly colored background (yellow/blue/gray) represents segmentation of the genome based on phasing.

Switches in segmentation from a mixed segment to a phased segment represent sister chromatid exchange events. The SCE (brown) and non-SCE (salmon) strand mutations can be inferred in mixed regions neighboring SCE sites. **c**, Metaplot of UV mutation density surrounding SCE sites. Shown are smoothed mutation rates/ megabase for 1 mb sliding windows with a 100 kb step size. Actual UV mutation density around SCE sites shown in black, while random selection of an equal number of mutations from other clones shown in red.

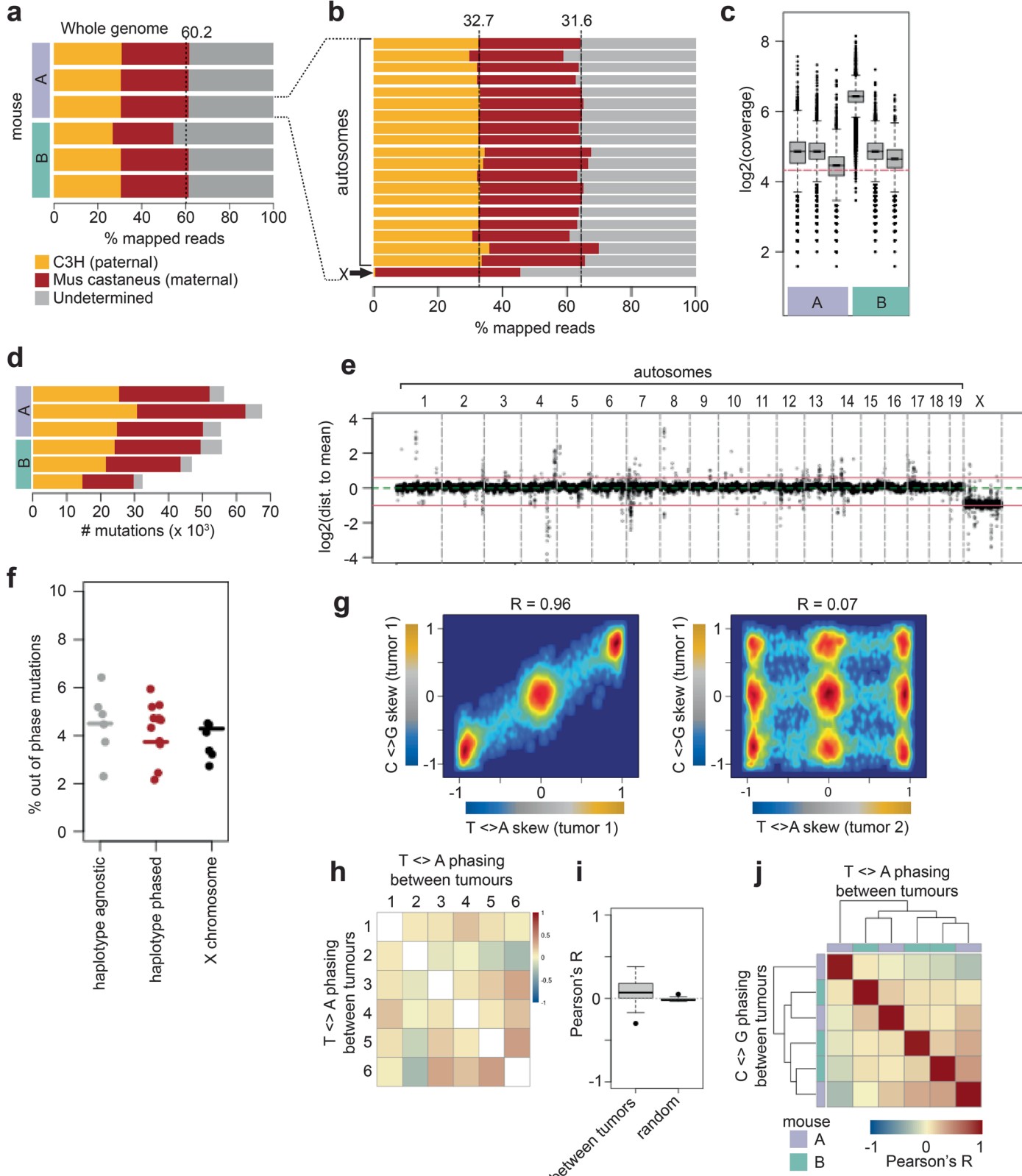

**Extended Data Fig. 7 | See next page for caption.**

**Extended Data Fig. 7 | Haplotype resolved lesion segregation in F1 mice.**
**a**, Percentage of mapped reads to a specific genome. Each bar represents a
tumor where brown depicts *Mus castaneus* specific reads and gold C3H. Mean
haplotype-specific reads for all 6 libraries is denoted by the vertical dashed line.
Colored boxes on the left denote which mouse the tumor was isolated from.
**b**, Chromosome specific mapping rate for one of the tumors in (a). Note *Mus
castaneus* specific mapping to the X. **c**, Read coverage per mutation in each
library. Horizontal red dashed line represents 20x coverage. Colored boxes below
denote mouse of origin as in (a). Boxplot elements are as described in Fig. 3c.
**d**, Number of mutations per tumor. Colors and symbols as in panel (a). **e**, Genomic
stability of F1 tumors. Read counts in 10 kb windows for a representative tumor.
Y axis represents log2 counts subtracted from the mean of all bins. Horizontal
green line represents no difference, while horizontal red lines depict duplication
or haploid content. **f**, Percent of out-of-phase mutations. The proportion of
mutations with a reference T or A base was calculated within changepoint derived
segments (see Methods). In the haplotype-agnostic analysis (gray points), out
of phase represents the mean amount of mutations with an identity opposite
to its segment. As an example, mutations at A bases when mutations within the

segment are predominantly at T bases (n = 6 tumor genomes). For segments to be
considered 'phased' in the haplotype-agnostic analysis, at least 80% of mutations
in that segment had to be of a single type. Haplotype resolved whole-genome
mutations with opposing identity are shown in red (n = 12, 6 genomes with 2
alleles each), and X chromosome localized mutations in black (n = 6). Horizontal
lines represent the mean of each set of points. **g**, Left: Correlation between
T < > A phasing and C < > G phasing in a single tumor within 10 mb windows.
C and T mutations (as well as G and A) share phase within a single tumor. Right:
comparison between T < > A phasing in tumor 2 to C < > G phasing in tumor 1.
**h**, Comparison of T < > A phasing across all 6 tumors, as measured by Pearson
correlation for phasing in 10 mb bins. **i**, Boxplot of correlation of T < > A phasing
between all non-self correlations (n = 15) as in (h), as well as random expectation
of phasing correlation seen after mixing bins for all 6 tumors (n = 15). The
solid middle line represents the median, the gray box depicts the interquartile
range between the 1st and 3rd quartile, while the whiskers define the minima and
maxima. Outliers are shown as points. **j**, Heatmap of T < > A phasing (x axis) with
C < > G phasing (y axis) between and within tumors. Mouse of origin are colored
as in (a).

Martin S. Taylor

# Reporting Summary

## Statistics

For all statistical analyses, confirm that the following items are present in the figure legend, table legend, main text, or Methods section.

| n/a | Confirmed | |
|---|---|---|
| ☐ | ☒ | The exact sample size (*n*) for each experimental group/condition, given as a discrete number and unit of measurement |
| ☐ | ☒ | A statement on whether measurements were taken from distinct samples or whether the same sample was measured repeatedly |
| ☐ | ☒ | The statistical test(s) used AND whether they are one- or two-sided *Only common tests should be described solely by name; describe more complex techniques in the Methods section.* |
| ☒ | ☐ | A description of all covariates tested |
| ☐ | ☒ | A description of any assumptions or corrections, such as tests of normality and adjustment for multiple comparisons |
| ☐ | ☒ | A full description of the statistical parameters including central tendency (e.g. means) or other basic estimates (e.g. regression coefficient) AND variation (e.g. standard deviation) or associated estimates of uncertainty (e.g. confidence intervals) |
| ☐ | ☒ | For null hypothesis testing, the test statistic (e.g. *F*, *t*, *r*) with confidence intervals, effect sizes, degrees of freedom and *P* value noted *Give P values as exact values whenever suitable.* |
| ☒ | ☐ | For Bayesian analysis, information on the choice of priors and Markov chain Monte Carlo settings |
| ☒ | ☐ | For hierarchical and complex designs, identification of the appropriate level for tests and full reporting of outcomes |
| ☐ | ☒ | Estimates of effect sizes (e.g. Cohen's *d*, Pearson's *r*), indicating how they were calculated |

*Our web collection on statistics for biologists contains articles on many of the points above.*

## Software and code

Policy information about availability of computer code

| Data collection | Data was collected using Illumina NovaSeq 6000 and NextSeq 2000 platforms |
|---|---|
| Data analysis | R scripts to reproduce all main figure panels can be downloaded from GitHub (https://github.com/odomlab2/Single-Mitosis-LSE) with DOI 10.5281/zenodo.10786189.<br><br>R packages used in analysis (R version 4.3.0)<br>Python 3.6.1<br>samtools 1.12<br>SNPsplit 0.4.0<br>TrimGalore 0.6.6<br>bowtie2 2.3.5.1<br>Strelka2 2.8.4<br>bcftools 1.10.2<br>gatk-tools 0.2.2<br>bedtools 2.24.0<br>macs2 2.1.2.1<br>Kallisto 0.46.0<br>GenomicRanges 1.52.1<br>Rsamtools 2.12.0<br>randtests 1.0.1<br>Repitools 1.46.0 |

```
changepoint 2.2.4
RColorBrewer 1.1-3
pheatmap 1.0.12
NMF 0.26
TxDb.Mmusculus.UCSC.mm10.knownGene 3.10.0
BSgenome.Mmusculus.UCSC.mm10 1.4.3
rtracklayer 1.60.0
ggplot2 3.4.4
vcfR 1.14.0
scales 1.2.1
regioneR 1.32.0
Gviz 1.44.2
bsub 1.1.2
viridis 0.6.4
QuasR 1.40.1
maptools 1.1-8
apcluster 1.4.11
mixtools 2.0.0
lsa 0.73.3
```

For manuscripts utilizing custom algorithms or software that are central to the research but not yet described in published literature, software must be made available to editors and reviewers. We strongly encourage code deposition in a community repository (e.g. GitHub). See the Nature Portfolio guidelines for submitting code & software for further information.

## Data

Policy information about availability of data

All manuscripts must include a data availability statement. This statement should provide the following information, where applicable:

- Accession codes, unique identifiers, or web links for publicly available datasets
- A description of any restrictions on data availability
- For clinical datasets or third party data, please ensure that the statement adheres to our policy

Fastq files for the WGS, RNA and ATAC-seq described here can be downloaded from Sequence Read Archive (SRA) under the accession number PRJNA934746. Processed files including mutation calls, TPM counts and ATAC peaks used in the analysis have been deposited in GEO under the accession GSE230579.

## Research involving human participants, their data, or biological material

Policy information about studies with human participants or human data. See also policy information about sex, gender (identity/presentation), and sexual orientation and race, ethnicity and racism.

| | |
|---|---|
| Reporting on sex and gender | N/A |
| Reporting on race, ethnicity, or other socially relevant groupings | N/A |
| Population characteristics | N/A |
| Recruitment | N/A |
| Ethics oversight | N/A |

Note that full information on the approval of the study protocol must also be provided in the manuscript.

## Field-specific reporting

Please select the one below that is the best fit for your research. If you are not sure, read the appropriate sections before making your selection.

☒ Life sciences  ☐ Behavioural & social sciences  ☐ Ecological, evolutionary & environmental sciences

For a reference copy of the document with all sections, see nature.com/documents/nr-reporting-summary-flat.pdf

## Life sciences study design

All studies must disclose on these points even when the disclosure is negative.

| | |
|---|---|
| Sample size | A minimum of 5 pairs was desired to assess reproducibility and increase total mutation numbers to study. 7 replicates were conducted in case some samples were problematic. No specific calculation was performed for mouse tumour selection. Instead, we sought to investigate 3 |

unique tumours from 2 independent mice (6 in total) to ascertain haplotype resolution reproducibility between both tumours and mice.

| | |
|---|---|
| Data exclusions | No data were excluded. |
| Replication | General mutation patterns were interrogated across all replicates, both in cell culture and tumours, and each figure displays points for each replicate for the corresponding contrast. |
| Randomization | Randomization of sister pairs and mice was not relevant as this is not a case-controlled study. |
| Blinding | Blinding was not relevant as these experiments were not a case-controlled study. |

# Reporting for specific materials, systems and methods

We require information from authors about some types of materials, experimental systems and methods used in many studies. Here, indicate whether each material, system or method listed is relevant to your study. If you are not sure if a list item applies to your research, read the appropriate section before selecting a response.

### Materials & experimental systems

| n/a | Involved in the study |
|---|---|
| ☒ | Antibodies |
| ☐ | ☒ Eukaryotic cell lines |
| ☒ | Palaeontology and archaeology |
| ☐ | ☒ Animals and other organisms |
| ☒ | Clinical data |
| ☒ | Dual use research of concern |
| ☒ | Plants |

### Methods

| n/a | Involved in the study |
|---|---|
| ☒ | ChIP-seq |
| ☐ | ☒ Flow cytometry |
| ☒ | MRI-based neuroimaging |

## Eukaryotic cell lines

Policy information about cell lines and Sex and Gender in Research

| | |
|---|---|
| Cell line source(s) | P388D1 cells from ATCC and Lentix HEK293T |
| Authentication | No authentication was carried out. |
| Mycoplasma contamination | Cell lines tested negative for mycoplasma contamination. |
| Commonly misidentified lines (See ICLAC register) | No commonly misidentified cell lines was used in this study. |

## Animals and other research organisms

Policy information about studies involving animals; ARRIVE guidelines recommended for reporting animal research, and Sex and Gender in Research

| | |
|---|---|
| Laboratory animals | Inbred female Mus musculus castaneus (CAST/EiJ) mice were crossed with inbred male C3H/HeOuJ (C3H) mice. The F1 offspring were treated with a single intraperitoneal dose of N-Nitrosodiethylamine (DEN; Sigma-Aldrich N0258; 20 mg/kg body weight) at P15. Liver tumours were isolated 30 weeks after treatment and stored at -80°C for DNA extraction and sequencing. Liver tissue from an untreated P15 litter mate was sampled for control experiments. Control samples (liver tissue) were also collected from untreated, age-matched littermates. Animals were maintained using standard husbandry: mice were group housed in Tecniplast GM500 IVC cages at room temperature/humidity with a 12 h:12 h light:dark cycle and ad libitum access to water, food (LabDiet 5058), and environmental enrichments. |
| Wild animals | This study did not use wild animals. |
| Reporting on sex | Only males were used, as females are relatively resistant to DEN induced tumorigenesis. |
| Field-collected samples | This study did not use field collected samples. |
| Ethics oversight | Animal experimentation was carried out in accordance with the Animals (Scientific Procedures) Act 1986 (United Kingdom) and with the approval of the Cancer Research UK Cambridge Institute Animal Welfare and Ethical Review Body (AWERB). |

Note that full information on the approval of the study protocol must also be provided in the manuscript.

# Flow Cytometry

## Plots

Confirm that:

☒ The axis labels state the marker and fluorochrome used (e.g. CD4-FITC).

☒ The axis scales are clearly visible. Include numbers along axes only for bottom left plot of group (a 'group' is an analysis of identical markers).

☒ All plots are contour plots with outliers or pseudocolor plots.

☒ A numerical value for number of cells or percentage (with statistics) is provided.

## Methodology

| | |
|---|---|
| Sample preparation | To ascertain DNA content, 2 drops of Hoechst 33342 Ready Flow™ Reagent from Invitrogen™ was added to 2 x 106 cells and placed in the incubator for 15 minutes. Cells were spun down, resuspended in Miltenyi Biotec FACs buffer and assayed using the BD FACSAria™ Fusion 3 system. Green fluorophores were ascertained with excitation at 488 nM and emission at 530 nM, while the orange fluorophore of G1 cells was excited at 561 nM and emission recorded at 586 nM. |
| Instrument | BD FACSAria Fusion 3 |
| Software | BD FACSDiva software 8.0.2 |
| Cell population abundance | Histogram of Hoechst staining of DNA content in supplemental figure 1c shows all signal for gated cells as depicted in supplemental gating figure. For single cell sorting to create the PF1 clone, positive Green fluorescence using signal as depicted in the supplemental gating figure was used. |
| Gating strategy | Live cells were determined by FSC-A and SSC-A, and singlets gated with FSC-A and FSC-H. Cells with positive emission in the 530/30 after excitation with a blue laser at 488nM were sorted as singlets for clonal selection. |

☒ Tick this box to confirm that a figure exemplifying the gating strategy is provided in the Supplementary Information.

