## [Peer Review File · Nature Genetics]

Peer Review Information

Manuscript Title: Single-mitosis dissection of acute and chronic DNA mutagenesis and repair

Corresponding author name(s): Professor Duncan Odom, Professor Martin Taylor

Reviewer Comments & Decisions:

Decision Letter, initial version:

30th Aug 2023

Dear Duncan,

First, please accept my sincere apologies for the delay in returning this decision to you. Thank you for your patience!

Your Article, "Single-mitosis dissection of acute and chronic DNA mutagenesis and repair" has now been seen by 3 referees. You will see from their comments below that while they find your work of interest, some important points are raised. We are interested in the possibility of publishing your study in Nature Genetics, but would like to consider your response to these concerns in the form of a revised manuscript before we make a final decision on publication.

We therefore invite you to revise your manuscript taking into account all reviewer Please highlight all changes in the manuscript text file. At this stage we will need you to upload a copy of the manuscript in MS Word .docx or similar editable format.

*2) If you have not done so already please begin to revise your manuscript so that it conforms to our Article format instructions, available

[here](http://www.nature.com/ng/authors/article_types/index.html).

*3) Include a revised version of any required Reporting Summary:

[redacted]

We are happy to be flexible about deadlines for receiving your resubmission (albeit with the usual disclaimers regarding novelty). That said, please let me know if you anticipate needing more than 6 months to complete the work.

Sincerely,

Safia Danovi
Editor
Nature Genetics

Referee expertise:

Referee #1: mutagenesis, cell division

Referee #2: DNA repair, genome instability

Referee #3: mutagenesis

Reviewers' Comments:

Reviewer #1:

Remarks to the Author:

Understanding how episodic exposure to mutagens generates patterns of mutations is fundamental to many human diseases. Prior to a 2020 paper from this group (Aitken et al. 2020 Nature), it was generally assumed that most acute DNA base lesions were repaired or fixed as base alterations within a single cell division cycle. Surprisingly, this 2020 paper concluded that lesions were often carried over beyond the first cell division resulting in (1) striking mutational strand bias and (2) subclonal heterogeneity from mutigenerational DNA repair. The footprint of this "lesion segregation model" of mutagenesis was then identified in human cancer genomes. The results are important for understanding how mutagens contribute to genetic variation and tumor development and potentially for making inferences about toxin exposure. Although the model was appealing, the evidence remained somewhat inferential because the analyzed tumors provided information about one daughter from the division of a damaged cell.

Here, Ginno et al. present an elegant single cell-derived analysis that enables both daughters (and subsequent progeny) to be tracked from the division of a damaged cell. The approach blends various existing technologies (e.g., Berkeley Lights microfluidic single cell isolation and FastFucci imaging) in an innovative way. The isolation of individual mitotic sister pairs following a single UV exposure enabled acute UV-induced damage (private to each daughter) to be clearly distinguished from long-accumulated ROS-mediated damage (shared by both daughters). Altogether, the work definitively establishes the lesion segregation model. In addition, the work adds to a debate about the role of transcription in base excision repair of oxidized guanines. The authors find no evidence of strand bias for oxidative damage, unlike UV damage, in transcribed regions. Instead, ATAC sequencing supported the idea that more efficient BER in transcribed regions is simply due to more open chromatin in these regions.

Major Point:

The authors hypothesize that persistence of unrepaired UV lesions may contribute to multi-allelic variation (i.e., replication across an unrepaired lesion over multiple cell cycles may generate multiple alleles in the population). To test this, the authors examine CpC dinucleotides, reasoning that these sites may form cyclobutane dimers which can lead to different mutational outcomes (CpC > TpT, CpC > CpT, CpC > TpC). Supporting their hypothesis, the authors observe multiple alleles at these sites within single lineages. While we understand the authors' reasoning, we have some technical concerns with the bioinformatic approach and interpretation.

When the authors examine mutations occurring at all 5'-CpC-3' in the genome, they observe that the 3' cytosine is more often mutated than the 5' cytosine (Figure 3b, c, and d, and main text). The

segregate, unrepaired over multiple cell generations, to cause chromosome-scale phasing of mutations. The authors further demonstrate that ROS mutations show random distribution between sister cells suggesting that gradual accumulation of mutations masks mutational asymmetry. ROS mutation rates are reduced in transcribed genes partially due to enhanced chromatin accessibility of BER factors. The authors confirm that lesion segregation and mutation phasing occur in an in vivo mouse model of liver cancer.

The results here are impactful, the experiments rigorously performed, and the methods and models are highly innovative. I am therefore unequivocally in support of publication. However, I do have several comments that I think should be addressed prior to publication.

Comments:

Lesion segregation model is an attractive model to explain mutational phasing. However, the authors didn't provide an explanation for why lesions are not repaired in the same cell cycle and what pathways may be involved in regulating this phenomenon. Also, more work is needed to determine the fraction of lesions that are unrepaired relative to the total number of induced lesions in sister cells.

Much of the acute DNA damage studies are limited to UV exposure. The authors should consider validating more key results using additional mutagens.

The authors show that PF1 cells exhibit comparable growth rates in the Berkeley lights chamber as in standard cell culture. They should test whether these results extend to other experimental conditions such as UV exposure. This is an important control as selection biases may be introduced here based on cell capacity to proliferate following UV exposure.

Regarding the rate of ROS mutations in transcribed genes, the authors should provide evidence of enhanced recruitment of BER factors in transcribed gene bodies due to enhanced chromatin accessibility to support their claim.

It was not clear how closely the mutation landscape observed in this system matches human cancer genomes and whether lesion segregation contributes to tumor evolution.

Reviewer #3:

Remarks to the Author:

This manuscript presents an impressive experimental platform for the analysis of damage induced mutations. The platform tracks sister cells following a single mitotic division. The authors also describe results of the analysis of F1 mice exposed to DEN mutagenesis. Both experimental approaches are innovative and very clever. My major concern with the manuscript is that the study did not generate any new observations. The manuscript confirms many highly important earlier findings (most published by some of the same authors). It seems that none of the reported results is truly enabled by the new technology. The results reported in Figures 3 and 6 are obtainable from analyzing single-cell derived clones, Figures 4 and 7 are not related to the new technology, and Figure 5 recapitulates an important conceptual finding published in Anderson et. al., ref. 15. In sum, the manuscript describes an interesting new technology, but its application mostly recapitulates known observations. Other specific comments are listed below.

- 1) I am wondering if the authors would consider more sophisticated techniques of statistical analysis or describe the current analyses in better detail. For example, it is not clear how was the difference between VAF distributions measured. As another example, the analysis of the ATAC-seq effect could be performed using a multivariate model incorporating other epigenetic covariates and the formal partitioning of variance explained.
- 2) A significant portion of the paper is focused on the analysis of background accumulation of C>A mutations. The authors postulate that these mutations are generated by ROS based on the similarity with Signature 18. Although it is possible that these are ROS mutations, there are a few other signatures that correlate well with the observed spectra (Supplementary Figure 2c). I suggest, at minimum, including a caveat when naming these mutations "ROS".
- 3) I am slightly puzzled that some of the independent clones share as much as 4.7% of mutations. Is there an intuitive explanation?
- 4) I suggest avoiding the term "mutation load" in the context of this study. It evokes the original Haldane's definition that is the relative fitness loss.
- 5) Data availability: VCFs will be shared in the future and R scripts are available upon request. Please check if it is consistent with the Nature Genetics policy.
- 6) It would be good to cite this study of multiallelic mutations in cancer
<https://www.nature.com/articles/s41588-021-01005-8>.

Author Rebuttal to Initial comments

**Response to comments for article NG-A62569**

*We would like to thank the reviewers for providing helpful feedback. We have made*
 *several changes outlined below regarding each suggestion and criticism and feel*
 *these modifications have greatly improved the manuscript.*

*In addition to reviewer specific changes, we have made the following additional*
 *corrections:*

- *1) Several R packages were not properly cited in the methods (randtests,*
 *Rsamtools, etc.) and we have now added these citations.*
- *2) In the mutation rate calculation section of methods, there was a typo on line*
 *1162. The 192 mutation types were folded into 64 categories, not 62.*
- *3) Extended Data Fig 4g was updated to reflect the same analysis of OGG1*
 *(Extended Data Fig 4h). This included using 5kb flanks instead of 10kb as this*
 *allowed for more comparable numbers between bins.*

**Reviewer 1:**

**Understanding how episodic exposure to mutagens generates patterns of**

mutations is fundamental to many human diseases. Prior to a 2020 paper from
this group (Aitken et al. 2020 Nature), it was generally assumed that most
acute DNA base lesions were repaired or fixed as base alterations within a
single cell division cycle. Surprisingly, this 2020 paper concluded that lesions
were often carried over beyond the first cell division resulting in (1) striking
mutational strand bias and (2) subclonal heterogeneity from multigenerational
DNA repair. The footprint of this “lesion segregation model” of mutagenesis
was then identified in human cancer genomes. The results are important for
understanding how mutagens contribute to genetic variation and tumor
development and potentially for making inferences about toxin exposure.
Although the model was appealing, the evidence remained somewhat
inferential because the analyzed tumors provided information about one
daughter from the division of a damaged cell.

Here, Ginno et al. present an elegant single cell-derived analysis that enables
both daughters (and subsequent progeny) to be tracked from the division of a
damaged cell. The approach blends various existing technologies (e.g.,
Berkeley Lights microfluidic single cell isolation and FastFucci imaging) in an
innovative way. The isolation of individual mitotic sister pairs following a
single UV exposure enabled acute UV-induced damage (private to each
daughter) to be clearly distinguished from long-accumulated ROS-mediated
damage (shared by both daughters). Altogether, the work definitively
establishes the lesion segregation model. In addition, the work adds to a
debate about the role of transcription in base excision repair of oxidized
guanines. The authors find no evidence of strand bias for oxidative damage,
unlike UV damage, in transcribed regions. Instead, ATAC sequencing
supported the idea that more efficient BER in transcribed regions is simply
due to more open chromatin in these regions.

*We thank the reviewer for this concise synopsis of our work and recognizing the key*
*findings of our study.*

**Major Points**

**When the authors examine mutations occurring at all 5'-CpC-3' in the genome,**
**they observe that the 3' cytosine is more often mutated than the 5' cytosine**
**(Figure 3b, c, and d, and main text). The authors conclude that the 3' cytosine**
**in a CpC dimer is inherently more prone to mutation. However, we believe that**
**there may be an alternative technical explanation. The identification of C > T**
**mutations within 5'-CpC-3' dinucleotides does not guarantee that this mutation**
**arose as the result of a CpC dimer. For instance, the same mutation could**
**arise from a CpT dimer if a thymine is immediately downstream. We can then**
**see that, considering all 5'-CpC-3' dinucleotides in the genome, it is sometimes**
**possible for the 3' cytosine to form a CpT dimer (whenever the downstream**
**base is T), whereas it is never possible for the 5' cytosine to form a CpT dimer**
**(because the following base is always a C). This caveat becomes even more**
**important when you consider the differing percentages of C > T mutations for**
**5'-CCT-3' versus 5'-TCC-3' motifs in COSMIC SBS7a UV-induced mutation**
**signature (below, referenced by the authors in Figure 2e). For these reasons,**
**we believe the frequencies reported in Figure 3b, c, and d may be misleading.**
**Because this point is not central to the conclusions of the manuscript, we**
**suggest that this point could be removed. Alternatively, the authors could**
**refine their analysis by grouping CpC dinucleotides according to their**
**sequence context (i.e. the bases on the 5' and 3' ends of the CpC dinucleotide).**

*This is a very insightful comment. We have included an additional analysis to*
*specifically address this point.*

*The reviewer is suggesting that although multi-allelic mutations are occurring in the*
*context of CC dinucleotides, the observed mutations may actually arise from other*
*pyrimidine dimers overlapping the CC; for example TCC or CCT (red indicating the*
*reference base at the mutated site, underline the pyrimidine dimer).*

*As the reviewer notes, the COSMIC SBS7a signature associated with UV damage*
*shows a higher frequency of single mutations in a TCC context than a CCT context.*

*As all multi-allelic sites in our study were identified based on the presence of CC->TT*
*dinucleotide mutations in addition to single nucleotide changes at the same site in*
*the same clone, this already implicates a CC dimer rather than overlapping*
*alternatives in the clone progenitor. The bias towards TCC over CCT mutations in*
*SBS7a would also suggest a higher frequency of CC->TC than CC->CT mutations*
*which is the opposite to our findings at multi-allelic sites (Fig. 3).*

*The features of COSMIC SBS7a noted by the reviewer do highlight that it is*
*important to consider the nucleotide context of the dimer, rather than just focussing*
*on the immediate context of individual mutated nucleotides. To further address this*
*point, we looked at the frequency of the 5 bases immediately upstream and*
*downstream of all multi-allelic sites. If a pyrimidine 3' of the CC base is driving the*
*phenomenon we describe, our expectation would be a higher proportion of either C*
*or T downstream of the multi-allelic sites when compared to biallelic mutations.*

*From this analysis we observe that there is no significant difference in base*
*distribution surrounding multi-allelic versus biallelic dual mutations. We reference this*
*analysis in section 3 (lines 257-262) with the following segment:*

*“To test whether a pyrimidine directly 3' to the CC site might explain the 3' bias, we*
*looked at base composition surrounding multi-allelic sites and compared it to both*
*biallelic and randomly sampled CC sites across the genome. On the contrary,*
*pyrimidines were significantly enriched directly 5' from CC dual mutations (Extended*
*Data Fig. 3h), arguing that the 3' bias is not caused by a dimer directly downstream*
*of multiallelic sites.”*

**Minor point: We note that P388D1 is a mouse cell line. The authors should**
**clarify whether their mutational signature comparison to COSMIC SBS**
**references (Figure 2e) was performed using frequencies that were normalized**
**for the sequence content of the mouse genome.**

*The signatures used in the analysis were downloaded from the COSMIC SBS*
*database, version 3.2 which were based on the mouse mm10 genome reference.*
*This was not explicitly stated in the manuscript and we have changed the methods*
*section titled 'Mutation Signatures' to remedy this (lines 1178 to 1179). We thank the*
*reviewer for noting the omission.*

**Reviewer 2:**

**In this manuscript the authors present the development of a new approach to**
**track mutations, introduced by induced or chronic mutational processes, over**
**single cell divisions. The innovative approach depends on microfluidics based**
**cell splitting after a controlled number of divisions. This method allows the**
**authors to distinguish between acute UV and chronic ROS mutagenesis. They**
**find that a subset of acute UV-induced lesions remain unrepaired for more**
**than one cell cycle. This phenomenon of retained UV damage across cell**
**division leads to mutational phasing and multi-allelic variation in CC**
**dinucleotides. Much of the data presented here support hypotheses arising**
**from an earlier publication from this same group (Aitken et al 2020) that first**
**proposed that DNA lesions segregate, unrepaired over multiple cell generations,**
**to cause chromosome-scale phasing of mutations. The authors further**
**demonstrate that ROS mutations show random distribution between sister**
**cells suggesting that gradual accumulation of mutations masks mutational**
**asymmetry. ROS mutation rates are reduced in transcribed genes partially due**
**to enhanced chromatin accessibility of BER factors. The authors confirm that**
**lesion segregation and mutation phasing occur in an in vivo mouse model of**
**liver cancer.**

**The results here are impactful, the experiments rigorously performed, and the**
**methods and models are highly innovative. I am therefore unequivocally in**
**support of publication. However, I do have several comments that I think**
**should be addressed prior to publication.**

*We thank the reviewer for highlighting the key novelties of our work and the*
*supportive remarks regarding our manuscript. We will save the penultimate two*
*sentences for a rainy day.*

**Major Points:**

**Lesion segregation model is an attractive model to explain mutational phasing.**
**However, the authors didn't provide an explanation for why lesions are not**
**repaired in the same cell cycle and what pathways may be involved in**
**regulating this phenomenon. Also, more work is needed to determine the**
**fraction of lesions that are unrepaired relative to the total number of induced**
**lesions in sister cells.**

*Upon rereading the discussion, we agree with the reviewer that this topic was not*
*adequately addressed. To remedy this, we have added the following paragraph to*
*the discussion (lines 604 to 616).*

*"The lesion segregation mechanism is defined by persistent DNA damage. While*
*eukaryotic cells clearly possess repair mechanisms to resolve these lesions¹,*
*replication over altered bases seems to be driving most of the mutational process for*
*acute genotoxins^{2,3}, regardless of transformation. Resulting genomic signatures of*
*these mutations are ultimately the result of damaged base identities and several*
*other factors⁴. While the precise reasons for why a particular damage event may*
*persist remain to be fully understood, several key factors have been implicated in*
*this process. These include chromatin accessibility⁵, repair pathway efficiency, total*
*damage events at any one time, transcriptional state⁶ and sequence context. One*
*key missing piece of information is assigning an accurate quantitative estimate to the*
*number of specific DNA lesions following a genotoxic event. Future studies will be*
*required to help bridge the gap in our understanding between the total extent of*
*damage following mutagenic exposure and the resulting mutational landscape."*

**Much of the acute DNA damage studies are limited to UV exposure. The**

**authors should consider validating more key results using additional**
**mutagens.**

*While we share the reviewer's opinion that ultimately it is desirable to undertake such*
*extensive profiling of all clinically relevant and experimentally useful mutagens, our*
*work here does describe 3 distinct mutagenic pressures in two disparate systems.*
*Our three chosen mutagenic insults include a major clinically relevant environmental*
*source of DNA damage (UV), oxidative damage (ROS) that is a ubiquitous source of*
*endogenous DNA damage enhanced by a variety of metabolic conditions and*
*environmental exposures, and alkylation damage (DEN) similar to that widely used*
*as a chemotherapeutic (e.g. temozolomide). A significant novelty of our study is the*
*description of a system that can provide flexibility in addressing future work where*
*distinct variables can be accounted for (mutagen, repair pathway, cell cycle stage,*
*division number etc.). We fully agree that there are many follow-on questions that*
*can be addressed by building on the novel systems and results presented in this*
*manuscript, but feel that this goes beyond the scope of the current study.*

**The authors show that PF1 cells exhibit comparable growth rates in the**
**Berkeley lights chamber as in standard cell culture. They should test whether**
**these results extend to other experimental conditions such as UV exposure.**
**This is an important control as selection biases may be introduced here based**
**on cell capacity to proliferate following UV exposure.**

*This was an oversight on our part in noting what is depicted in the proliferation data*
*shown in Extended Data Figure 1a. The pen measurements from Berkeley lights*
*used to calculate these numbers were from cells treated with UV that proliferated*
*post treatment. We have updated the panel and legend for Extended Data Fig. 1a to*
*account for this.*

*Furthermore, in consideration of this comment we reassessed both our method*
*description of the division time estimates as well as the rates themselves. In the*
*original submission, we had calculated doubling times between each consecutive*
*time point, which led to a larger spread of values driven by a few outlier data points.*

*To incorporate all measurements of a pen into a single rate measurement, we fit a*
*linear model regressing log₂(cell counts) on time in hours of culture on the machine.*
*We have also now distinguished between pens that were split as opposed to those*
*that were not split in Extended Data Figure 1a.*

*The results of this alternate rate assignment did not impact our original statement in*
*the manuscript section 1, lines 112-113: “Under optimized culture conditions (see*
*methods), cells divide at a rate comparable to that measured in standard cell culture*
*(Extended Data Fig. 1a)”. However, we have updated the methods section entitled*
*“Cell culture and splitting on the Berkeley Lights Lightning platform” (line 863-871) to*
*more explicitly state how these rate calculations were made.*

**Regarding the rate of ROS mutations in transcribed genes, the authors should**
**provide evidence of enhanced recruitment of BER factors in transcribed gene**
**bodies due to enhanced chromatin accessibility to support their claim.**

*Our findings indeed suggest that repair factors capable of recognizing oxidative DNA*
*damage should be recruited to highly expressed genes. To address this, we looked*
*at the genomic occupancy of the oxidative guanine repair protein OGG1 in HEK293*
*cells (PMID: 29940424, GEO Accession: GSE89017). We compared this to*
*ribosomal RNA depleted total RNA from HEK293 cells (PMID: 27852650, GEO*
*Accession: GSE76496) to determine how transcriptional output is reflected in*
*recruitment of OGG1 to gene bodies.*

*We profiled OGG1 ChIP-seq signal across gene bodies exactly as described for*
*ATAC-seq after binning genes based on transcriptional levels. This analysis*
*recapitulated increased OGG1 recruitment to active gene promoters as originally*
*described⁷. Furthermore, gene body OGG1 recruitment increased with transcriptional*
*output (Extended Data Panel 4h). We have now added panel h to Extended Data*
*Figure 4 and this analysis is referenced on lines 324-326. We have also included a*
*section in methods entitled “HEK293 OGG1 ChIP and total RNA-seq processing and*
*analysis” to describe how this was done (lines 1034-1052).*

**It was not clear how closely the mutation landscape observed in this system**
**matches human cancer genomes and whether lesion segregation contributes**
**to tumor evolution.**

*To address this concern, we have revised the 2nd paragraph of our Discussion*
*adding the lines (566-573) below:*

*“our mechanistic insights are relevant to human cancers because: (1) similar*
*phased mutational profiles are seen in patient data^{14,52}, (2) the mutation signatures in*
*our system are incredibly similar to relevant tumour tissues (e.g. skin and UV), (3)*
*we recapitulate reciprocal mutation phasing of the ‘lost sister’ predicted at*
*transformation and clonal expansion. In addition, we also demonstrate that multi-*
*allelic variation seen in hundreds of mouse tumours is observable in a non-*
*cancerous setting. These findings further underscore the major role lesion*
*segregation plays in establishing genomic mutational patterns observed in cancer.”*

**Reviewer 3:**

**This manuscript presents an impressive experimental platform for the analysis**
**of damage induced mutations. The platform tracks sister cells following a**
**single mitotic division. The authors also describe results of the analysis of F1**
**mice exposed to DEN mutagenesis. Both experimental approaches are**
**innovative and very clever. My major concern with the manuscript is that the**
**study did not generate any new observations. The manuscript confirms many**
**highly important earlier findings (most published by some of the same**
**authors). It seems that none of the reported results is truly enabled by the new**
**technology. The results reported in Figures 3 and 6 are obtainable from**
**analyzing single-cell derived clones, Figures 4 and 7 are not related to the new**
**technology, and Figure 5 recapitulates an important conceptual finding**
**published in Anderson et. al., ref. 15. In sum, the manuscript describes an**
**interesting new technology, but its application mostly recapitulates known**
**observations. Other specific comments are listed below.**

*We thank the reviewer for their supportive remarks, and below we address the*
*reviewer's specific concerns in detail. The major impacts of our study are*
*mechanistic hypothesis testing.*

*Most importantly, our adaptation of this microfluidics platform to precisely track single*
*mitotic events was absolutely essential to conclusively demonstrate the previously-*
*inferred mechanism underlying lesion segregation. The model of lesion segregation*
*was in large part inferential as a critical missing component was the lost sister cell at*
*the time of transformation. Rigorously testing this assumption necessitated a*
*platform controlling for (1) a single mitotic division (2) cell cycle assessment and (3)*
*mutagen exposure. We have now reworded the introduction on lines 84-86 to more*
*explicitly state this novelty of our work.*

*We demonstrate for the first time a previously unrecognized mode of multi-allelic*
*variation: the generation for both di-nucleotide and mono-nucleotide substitutions at*
*the same site from repeated replication over a persistent lesion. In Figure 3, it is true*
*that multi-allelism could have been observed using single-cell derived clones.*
*However, microfluidic isolation of daughter cells enabled us to increase our detection*
*sensitivity two-fold by necessitating at least one cell division after treatment. This*
*increase in detection sensitivity was crucial for the novelty of Figure 3 in analyzing 3'*
*bias at multi-allelic sites because of the relatively rare nature of these events.*

*In Figure 4, the sister separation resulting from using the microfluidics system*
*directly enabled us to distinguish between two separate mutagenic pressures*
*occurring concurrently. Conclusively separating these mutational processes is*
*otherwise not possible, as sister shared mutations have exclusively the ROS*
*mutational signature. This allowed us to explore how accessibility and transcription*
*are affecting distinct mutation distributions and provide clarity to an ongoing debate*
*regarding transcription associated repair of ROS damage.*

*In Figure 5, the mutation phasing phenomenon is indeed anchored in concepts*
*presented previously^{2,3}. In our current submission, however, it is used primarily as a*

*basis to demonstrate how chronic low mutation pressure can mask phasing*
*observed for acute bursts, with both occurring in the same cell. These specific*
*results demonstrate how chronic damage could mask a phased mutation phenotype*
*in human cancer evolution and this was enabled from the single mitotic splitting by*
*allowing us to discriminate between mutation types.*

*In Figure 6, mirror-image mutational phasing fundamentally requires unbroken*
*tracking of the two sisters from a single mitotic division. Comparison of 371 liver*
*tumors from mice demonstrate that the phasing patterns for each tumor are*
*independent events², underscoring how our system enabled this discovery.*

*In Figure 7, the figure's purpose is to reveal an additional line of in vivo evidence to*
*support the lesion segregation model; as such, this figure does not use the*
*microfluidics system. By haplotype resolution of mutations, we demonstrate that*
*phasing across all chromosomes is absolute, and that ~95% of all T > N mutations*
*are in agreement with their phase, a number that could only be extracted from the X*
*chromosome from existing tumor data. This analysis also unequivocally*
*demonstrates for the first time that intrachromosomal transitions in mutation*
*asymmetry are sister chromatid exchange events.*

*Here, we present a highly controlled study that reduces as many variables as*
*possible to enable (1) interrogation of the mechanism of lesion segregation and (2)*
*discriminate between co-occurring mutational processes.*

**I am wondering if the authors would consider more sophisticated techniques**
**of statistical analysis or describe the current analyses in better detail. For**
**example, it is not clear how was the difference between VAF distributions**
**measured.**

*To expand and clarify the details regarding comparison of VAF distributions in figure*
*2, we have calculated Pearson's median coefficient of skewness. This has been*
*added as panel i in figure 2, and shows that sister shared SNVs reflect very little*

*skew in the VAF distribution, while UV and ROS reveal increasingly positive skew.*
*This is the expected outcome given the shift in the VAF distribution to lower values*
*for UV and more so for ROS. Additionally we reference this panel in the text of*
*section 2 on lines 188-192.*

**As another example, the analysis of the ATAC-seq effect could be performed**
**using a multivariate model incorporating other epigenetic covariates and the**
**formal partitioning of variance explained.**

*Our interpretation is that the reviewer is suggesting a modeling analysis combining*
*our ATAC data with additional epigenetic profiling experiments. The goal of such an*
*analysis would be to dissect how chromatin state, in addition to transcription status,*
*can influence repair rates.*

*While public data is available for B-Cell and erythroid progenitors, our sample-*
*matched and highly-reproducible current measurements (RNA-seq, ATAC-seq,*
*WGS) are from a monocyte/macrophage line (P388D1). This would additionally*
*require clarity on what should be measured (histone marks, DNA*
*(hydroxy)methylation, chromatin structure, etc.). We would argue such an*
*exploratory analysis would require numerous additional profiling experiments in our*
*line to be informative and goes well beyond the scope of this work.*

*Recognizing the importance of connecting other covariates, however, we looked at*
*the genomic occupancy of the oxidative guanine repair protein OGG1 in HEK293*
*cells. We compared this to cell-type matched ribosomal RNA depleted total RNA to*
*determine how transcriptional output is reflected in recruitment of OGG1 to gene*
*bodies. This demonstrates that OGG1 gene body signal increases from low/non-*
*transcribed gene bodies to highly transcribed gene bodies (Extended Data Fig 4h), in*
*agreement with our mutation rate findings for ROS.*

*In sum, our original ATAC-seq data as well as the added OGG1 analysis argue that*
*chromatin accessibility plays a strong role in ROS repair.*

**A significant portion of the paper is focused on the analysis of background**
**accumulation of C>A mutations. The authors postulate that these mutations**
**are generated by ROS based on the similarity with Signature 18. Although it is**
**possible that these are ROS mutations, there are a few other signatures that**
**correlate well with the observed spectra (Supplementary Figure 2c). I suggest,**
**at minimum, including a caveat when naming these mutations “ROS”.**

*We have modified the 4th paragraph of section 2 (line 176-179) to indicate this*
*caveat:*

*“...sharing a signature most similar to SBS18⁸ and suggesting they are ROS induced*
*mutations (Fig. 2d middle, Extended Data Fig. 2c). This is in agreement with*
*previous work profiling mutations influenced by high oxygen conditions in cell*
*culture⁹ and 8-oxo-G in human dermal fibroblasts¹⁰.”*

**I am slightly puzzled that some of the independent clones share as much as**
**4.7% of mutations. Is there an intuitive explanation?**

*Yes, these shared mutations are most likely due to clonally shared ancestry.*

*To add clarity, we have included an additional panel in Extended Data Figure 2e,*
*showing the overlap of mutations between clones (sisters are grayed out in the*
*middle to stress clonal sharing). The 4 sister genomes with increased clonal*
*mutation sharing (Figure 2c, bottom four points between 4.3 and 4.7%) stem from*
*two individual clones: D and F (Extended Data Fig 2e). These 2 clones with more*
*shared mutations were penned in one of the two Berkeley Light chips used in our*
*study (annotated in Extended Data Fig. 2e), which were conducted 5 months apart.*
*Our interpretation is that additional mutations accrued and subselection of the*

*population through later passages led to this phenomenon.*

*Our interpretation here is that these two cells came from a more recent common*
*ancestor, and thus share 227 mutations with 85% of these mutations also shared*
*between sisters. This is in stark contrast to the 0-31 mutations shared across other*
*clones (0.33 to 1.2%). We have included the following sentence in section 2 (line*
*192-197) to reference this panel:*

*“This ancestral hypothesis is supported by the observation that the four sister*
*genomes with higher clonal mutation sharing (Fig 2c, bottom) are derived from two*
*singly penned clones on the same Berkeley lights chip, and thus likely share a more*
*recent common ancestor (Extended Data Fig. 2e) in agreement with previous*
*findings in mammalian cell culture²³.”*

**I suggest avoiding the term “mutation load” in the context of this study. It**
**evokes the original Haldane’s definition that is the relative fitness loss.**

*We have changed references of “mutation load” to either “mutations”, “total*
*mutations”, “mutation density”, “mutation rate” or “total damage” where*
*appropriate.*

**Data availability: VCFs will be shared in the future and R scripts are available**
**upon request. Please check if it is consistent with the Nature Genetics policy.**

*We have closely followed the Nature Genetics policy on data and tool*
*availability.*

423 *Our Data availability statement in the manuscript and in the reporting summary*
*is:*

*“Fastq files for the WGS, RNA and ATAC-seq produced for this manuscript can be*

*downloaded from Sequence Read Archive (SRA) under the accession number*
*PRJNA934746. Processed files including mutation calls, TPM counts and ATAC*
*peaks used in the analysis have been deposited in GEO under the accession*
*GSE230579. HEK293 Flag-OGG1 ChIP-seq data was downloaded from the GEO*
*accession GSE89017 while HEK293 Ribo-Zero total RNA was obtained from GEO*
*accession GSE76496. Custom R scripts used for analysis are available upon*
*request. Reviewer access:*
*SRA: <https://dataview.ncbi.nlm.nih.gov/object/PRJNA934746?reviewer=2q34dbljblta1qcqr19obutt76>*
*GEO: <https://www.ncbi.nlm.nih.gov/geo/query/acc.cgi?acc=GSE230579>*
*GEO token: odopmccwjlglir”*

**It would be good to cite this study of multiallelic mutations in cancer**

[https://www.nature.com/articles/s41588-021-01005-8.](https://www.nature.com/articles/s41588-021-01005-8)

*In the above reference, there is one instance of the term ‘multi-allelic’ in the*
*discussion and is focused on the biallelic (or homozygous) mutations in cancer*
*genomes. In the requested citation, biallelic refers to the same mutation on both*
*alleles in a tumor, or perhaps more simply mutation homozygosity. Because this*
*could lead to ambiguity regarding our slightly different definition of biallelic mutations,*
*the citation has been introduced to lines 236-237 of the manuscript:*

*“This observation is distinct from biallelic mutations seen in melanoma where*
*both*
*haplotypes are mutated²⁶”*

**References**

**1.** Huang, R. & Zhou, P.-K. DNA damage repair: historical perspectives, mechanistic

- pathways and clinical translation for targeted cancer therapy. *Signal Transduct Target*
*Ther* **6**, 254 (2021).
- 2. Aitken, S. J. *et al.* Pervasive lesion segregation shapes cancer genome evolution.
*Nature* **583**, 265–270 (2020).
- 3. Anderson, C. J. *et al.* Strand-resolved mutagenicity of DNA damage and repair. *bioRxiv*
2022.06.10.495644 (2022) doi:10.1101/2022.06.10.495644.
- 4. Hauer, M. H. & Gasser, S. M. Chromatin and nucleosome dynamics in DNA damage
and repair. *Genes Dev.* **31**, 2204–2221 (2017).
- 5. Bhakat, K. K., Mokkalapati, S. K., Boldogh, I., Hazra, T. K. & Mitra, S. Acetylation of
human 8-oxoguanine-DNA glycosylase by p300 and its role in 8-oxoguanine repair in
vivo. *Mol. Cell. Biol.* **26**, 1654–1665 (2006).
- 6. Menoni, H., Di Mascio, P., Cadet, J., Dimitrov, S. & Angelov, D. Chromatin associated
mechanisms in base excision repair - nucleosome remodeling and DNA transcription,
two key players. *Free Radic. Biol. Med.* **107**, 159–169 (2017).
- 7. Hao, W. *et al.* Effects of the stimuli-dependent enrichment of 8-oxoguanine DNA
glycosylase1 on chromatinized DNA. *Redox Biol* **18**, 43–53 (2018).
- 8. Tate, J. G. *et al.* COSMIC: the Catalogue Of Somatic Mutations In Cancer. *Nucleic*
*Acids Res.* **47**, D941–D947 (2019).
- 9. Kucab, J. E. *et al.* A Compendium of Mutational Signatures of Environmental Agents.
*Cell* **177**, 821–836.e16 (2019).

- 10. Jin, S.-G., Meng, Y., Johnson, J., Szabó, P. E. & Pfeifer, G. P. Concordance of
hydrogen peroxide–induced 8-oxo-guanine patterns with two cancer mutation
signatures of upper GI tract tumors. *Science Advances* **8**, eabn3815 (2022).
- 11. Demeulemeester, J., Dentre, S. C., Gerstung, M. & Van Loo, P. Biallelic mutations in
cancer genomes reveal local mutational determinants. *Nat. Genet.* **54**, 128–133 (2022).

Decision Letter, first revision:

1st Dec 2023

Dear Dr Odom,

Thank you for submitting your revised manuscript "Single-mitosis dissection of acute and chronic DNA mutagenesis and repair" (NG-A62569R). It has now been seen by the original referees and their comments are below. The reviewers find that the paper has improved in revision, and therefore we'll be happy in principle to publish it in Nature Genetics, pending minor revisions to satisfy the referees' final requests and to comply with our editorial and formatting guidelines.

Sincerely,

Safia Danovi
Editor
Nature Genetics

Reviewer #1 (Remarks to the Author):

The authors have performed additional analysis that addresses our major concerns with the interpretation of multiallelic sites in CpC motifs. We do have one small issue that we suggest that the authors include as a discussion point. We think that the authors should mention that, formally, this analysis does not distinguish between mutations from adjacent NpCpCpN dimers versus a single NpCpCpN dimer. In fact, the overrepresentation of 5' pyrimidines at mutated CpC's suggests that dimers with flanking bases may contribute to the mutations shown in Fig. 4.

This caveat aside, we fully support publication in Nature Genetics in its current form.

David Pellman and Gregory Brunette

Reviewer #2 (Remarks to the Author):

The authors have satisfactorily addressed my prior comments through additional experimentation and analysis as well as new discussion points. I agree with their assessment that the suggestion for incorporating additional DNA damage models is outside of the scope of this current manuscript. I am therefore supportive of publication of this revised manuscript.

Reviewer #3 (Remarks to the Author):

The revised manuscript has not been substantially changed upon revision. It presents an interesting new technology and reports a direct observation of lesion segregation. Lesion segregation is not a new phenomenon, but seeing a direct evidence is important and comforting. There is a new analysis of di-nucleotide and mono-nucleotide changes at the same site (an interesting although less significant observation). The revised manuscript remains structures around findings on ROS mutagenesis and the importance of new technology to separate ROS and UV mutations. I am still unsure about the novelty of the findings and about the need of the new technology to analyze ROS mutagenesis. First, ROS is a highly plausible cause of the early shared mutations, but there is no evidence to support this hypothesis. Many publications indeed assume that SBS18 is caused by ROS, although there is no direct evidence of that. The similarity to SBS18 is not a proof that ROS is the cause of the observed mutations. In my opinion, it would be better to state the ROS origin of mutations as a hypothesis. More importantly, it is unclear why do we need to separate UV induced mutations from the earlier accumulated mutations at the single cell level to study properties of the early mutations. The manuscript would be improved if re-framed as description of technology and the direct observation of lesion segregation as the main story line with the additional analyses presented as secondary or omitted.

Author Rebuttal, first revision:

The authors have performed additional analysis that addresses our major concerns with the interpretation of multiallelic sites in CpC motifs. We do have one small issue that we suggest that the authors include as a discussion point. We think that the authors should mention that, formally, this analysis does not distinguish between mutations from adjacent NpCpCpN dimers versus a single NpCpCpN dimer. In fact, the overrepresentation of 5' pyrimidines at mutated CpC's suggests that dimers with flanking bases may contribute to the mutations shown in Fig. 4.

This caveat aside, we fully support publication in Nature Genetics in its current form.

David Pellman and Gregory Brunette

To address this point, we have added this penultimate sentence to section 3 of the manuscript results section:

"While this analysis suggests that the 3' mutation bias results from a single dimer between adjacent CC bases, we cannot completely rule out that an additional dimer in an NpCpCpN context may influence the mutation bias."

Final Decision Letter:

8th Mar 2024

Dear Duncan,

I am delighted to say that your manuscript "Single-mitosis dissection of acute and chronic DNA mutagenesis and repair" has been accepted for publication in an upcoming issue of Nature Genetics.

Your paper will be published online after we receive your corrections and will appear in print in the next available issue. You can find out your date of online publication by contacting the Nature Press Office (press@nature.com) after sending your e-proof corrections.

Please note that *Nature Genetics* is a Transformative Journal (TJ). Authors may publish their research with us through the traditional subscription access route or make their paper immediately open access through payment of an article-processing charge (APC). Authors will not be required to make a final decision about access to their article until it has been accepted. Find out more about Transformative Journals

Authors may need to take specific actions to achieve compliance with funder and institutional open access mandates. If your research is supported by a funder that requires immediate open access (e.g. according to Plan S principles) then you should select the gold OA route, and we will direct you to the compliant route where possible. For authors selecting the subscription publication route, the journal's standard licensing terms will need to be accepted, including [a href="https://www.nature.com/nature-portfolio/editorial-policies/self-archiving-and-license-to-publish"](https://www.nature.com/nature-portfolio/editorial-policies/self-archiving-and-license-to-publish). Those licensing terms will supersede any other terms that the author or any third party may assert apply to any version of the manuscript.

If you have not already done so, we invite you to upload the step-by-step protocols used in this manuscript to the Protocols Exchange, part of our on-line web resource, natureprotocols.com. If you complete the upload by the time you receive your manuscript proofs, we can insert links in your article that lead directly to the protocol details. Your protocol will be made freely available upon publication of your paper. By participating in natureprotocols.com, you are enabling researchers to more readily reproduce or adapt the methodology you use. [Natureprotocols.com](https://natureprotocols.com) is fully searchable, providing your protocols and paper with increased utility and visibility. Please submit your protocol to <https://protocolexchange.researchsquare.com/>. After entering your nature.com username and password you will need to enter your manuscript number (NG-A62569R1). Further information can be found at <https://www.nature.com/nature-portfolio/editorial-policies/reporting-standards#protocols>

Sincerely,

Safia Danovi, PhD
Senior Editor, Nature Genetics
ORCID: 0009-0007-7822-5479